# Evolution and diversity of biomineralized columnar architecture in early Cambrian phosphatic-shelled brachiopods

**Zhiliang Zhang[1,2]\*, Zhifei Zhang[3], Lars Holmer[4], Timothy P Topper[3,5], Bing Pan[1], Guoxiang Li[1]**

[1]State Key Laboratory of Palaeobiology and Stratigraphy, Nanjing Institute of Geology and Palaeontology, Chinese Academy of Sciences, Nanjing, China; [2]School of Natural Sciences, Macquarie University, Macquarie Park, Australia; [3]State Key Laboratory of Continental Dynamics, Shaanxi Key Laboratory of Early Life & Environments, Department of Geology, Northwest University, Xi'an, China; [4]Institute of Earth Sciences, Palaeobiology, Uppsala University, Uppsala, Sweden; [5]Department of Palaeobiology, Swedish Museum of Natural History Stockholm, Stockholm, Sweden

**\*For correspondence:**
zhiliang.zhang@nigpas.ac.cn;
zhangtensor@hotmail.com

**Competing interest:** The authors declare that no competing interests exist.

**Abstract** Biologically-controlled mineralization producing organic-inorganic composites (hard skeletons) by metazoan biomineralizers has been an evolutionary innovation since the earliest Cambrian. Among them, linguliform brachiopods are one of the key invertebrates that secrete calcium phosphate minerals to build their shells. One of the most distinct shell structures is the organo-phosphatic cylindrical column exclusive to phosphatic-shelled brachiopods, including both crown and stem groups. However, the complexity, diversity, and biomineralization processes of these microscopic columns are far from clear in brachiopod ancestors. Here, exquisitely well-preserved columnar shell ultrastructures are reported for the first time in the earliest eoobolids *Latusobolus xiaoyangbaensis* gen. et sp. nov. and *Eoobolus acutulus* sp. nov. from the Cambrian Series 2 Shuijingtuo Formation of South China. The hierarchical shell architectures, epithelial cell moulds, and the shape and size of cylindrical columns are scrutinised in these new species. Their calcium phosphate-based biomineralized shells are mainly composed of stacked sandwich columnar units. The secretion and construction of the stacked sandwich model of columnar architecture, which played a significant role in the evolution of linguliforms, is highly biologically controlled and organic-matrix mediated. Furthermore, a continuous transformation of anatomic features resulting from the growth of diverse columnar shells is revealed between Eoobolidae, Lingulellotretidae, and Acrotretida, shedding new light on the evolutionary growth and adaptive innovation of biomineralized columnar architecture among early phosphatic-shelled brachiopods during the Cambrian explosion.

## eLife assessment

This **valuable** study examines the evolution of the pillars in the shell architecture of organo-phosphatic brachiopods. The phylogenetic implications of this shell structure in relation to other early Cambrian brachiopod families are interpreted based on **solid** evidence. As such, this paper with interesting ideas regarding the evolution of brachiopod shell structure contributes to our understanding of the ecology and evolution of brachiopods as a whole.

## Introduction

The early Cambrian witnessed a great burst in diversity of animal body plans and biomineralized shell architectures around half a billion years ago (*Briggs, 2015*; *Erwin, 2015*; *Erwin, 2020*; *Budd and Jackson, 2016*; *Murdock, 2020*; *Yun et al., 2021*; *Zhang and Shu, 2021*; *Zhang et al., 2021c*). The novel process of biologically-controlled mineralization producing organic-inorganic composites (hard skeletons) in complex animals had played a vital role in the survival and fitness of early clades (*Balthasar, 2009*; *Cuif et al., 2010*; *Li et al., 2022*; *Skovsted et al., 2008*; *Yun et al., 2022*), and in turn built the fundamental blocks of complex marine ecosystems (*Bicknell and Paterson, 2018*; *Buatois et al., 2020*; *Chen et al., 2022*; *Zhang et al., 2010*; *Zhang et al., 2020a*). Since the early Cambrian, this adaptive evolution has been demonstrated and continuously preserved in brachiopods, one of the key members of the Cambrian Evolutionary Fauna (*Carlson, 2016*; *Harper et al., 2021*; *Harper et al., 2017*; *Sepkoski, 1984*). Among them, the phosphatic-shelled brachiopods are some of the most common animals in early Cambrian faunas (*Harper et al., 2017*). With a high fidelity of preservation and high abundance of biomineralized shells in the fossil record, their morphological disparity, diversity of shell structures and growth patterns, together with ecological complexity are preserved in great detail (*Chen et al., 2021*; *Claybourn et al., 2020*; *Topper et al., 2018*; *Zhang, 2018*; *Zhang et al., 2020b*; *Zhang et al., 2021b*).

Studying the processes by which organisms form biomineral materials has been a focus at the interface between earth and life sciences (*Williams et al., 1986*; *Roda and Mar, 2021*). Brachiopods are unique animals in having the ability to secrete two different minerals, calcium phosphate and calcium carbonate, making them the ideal group to further explore the processes of biomineralization (*Simonet Roda et al., 2022*; *Williams, 1997a*). Hard tissues composed of calcium phosphate with an organic matrix are also largely present in vertebrates, which have remarkably shaped the evolutionary trajectory of life on Earth (*Ruben and Bennett, 1987*; *Wood and Zhuravlev, 2012*). The origin of phosphate biomineralization in the evolutionary distant invertebrate brachiopods and vertebrates is still a big mystery in animal evolution (*Luo et al., 2015*; *Neary et al., 2011*; *Roda and Mar, 2021*). Thus, more research is needed in order to understand how to relate hierarchical structure to function in the very early examples of calcium phosphate-based biomineralization processes (*Kallaste et al., 2004*; *Weiner, 2008*). It is noteworthy that South China has been considered as one of centres for the origination and early dispersal of phosphatic-shelled brachiopods (*Zhang et al., 2021a*), and hence, it provides a great opportunity to explore the unique biomineralization process and consequent adaptive evolution of their earliest representatives during the Cambrian radiation.

The shell-forming process of brachiopods, although critical to understanding their poorly resolved phylogeny and early evolution, has long been problematic (*Carlson, 2016*; *Cusack and Williams, 2007*; *Holmer et al., 2008a*; *Murdock, 2020*; *Streng et al., 2007*; *Temereva, 2022*; *Williams et al., 2004*; *Williams and Cusack, 1999*). The epithelial cells of the outer mantle lobes have been considered to be responsible for the shell ornamentation and fabrics of brachiopods (*Williams and Cusack, 1999*). However, the biologically controlled process of brachiopod shell secretion at the cellular level is still unclear, although organic substrates are observed to be available for biomineral deposition during mantle activity (*Williams et al., 1997b*). Extensive studies have been conducted on living and fossil shells, but most of them are focused on articulated or carbonate-shelled representatives (*Cusack et al., 2010*; *Griesshaber et al., 2007*; *Simonet Roda et al., 2022*; *Roda and Mar, 2021*; *Ye et al., 2021*). By contrast, the linguliform brachiopods with shells composed of an organic matrix and apatite minerals that show extremely intricate architectures and permit exquisite preservation are less studied. The shell structural complexity and diversity, especially of their fossil representatives require further investigation (*Butler et al., 2015*; *Cusack et al., 1999*; *Streng et al., 2007*; *Williams and Holmer, 1992*; *Zhang et al., 2017*). The building of shells by microscopic cylindrical columns is a unique feature, which is restricted to the phosphatic-shelled brachiopods and their assumed ancestors (*Butler et al., 2015*; *Holmer et al., 2002*; *Holmer et al., 2008b*; *Holmer, 1989*; *Skovsted and Holmer, 2003*; *Williams and Holmer, 2002*). This type of columnar shell was previously believed to be exclusively restricted to micromorphic acrotretide brachiopods, a group that demonstrates more complex hierarchical architectures and graded structures compared to simple lamella shell structure in older lingulides (*Holmer, 1989*; *Williams and Holmer, 1992*). Further studies, however, reveal that diverse columnar shell architectures occur in other brachiopod groups, including stem group taxa, such as *Mickwitzia*, *Setatella*, and *Micrina* (*Butler et al., 2015*; *Holmer et al., 2008a*; *Skovsted et al.,*

*2010*; *Skovsted and Holmer, 2003*; *Williams and Holmer, 2002*), the lingulellotretid *Lingulellotreta* (*Holmer et al., 2008b*), the eoobolid *Eoobolus* (*Zhang et al., 2021a*), and the enigmatic *Bistramia* (*Holmer et al., 2008b*). However, the columnar architectures among the oldest linguliforms and their evolutionary variations have not been studied in detail. Different conditions of fossil preservation with varied taphonomic histories have compounded this issue as different depositional environments, pore-water geochemistry, and subsequent diagenetic and tectonic alteration often obscure the finer details of shell structures (*Butler et al., 2015*; *Holmer et al., 2008a*; *Streng et al., 2007*; *Ushatinskaya and Korovnikov, 2014*; *Zhang, 2018*).

For the first time, exquisitely well-preserved columnar shell structures are described here from the oldest known eoobolid brachiopods. *Latusobolus xiaoyangbaensis* gen. et sp. nov. and *Eoobolus acutulus* sp. nov. are reported, based on new specimens from the Cambrian Series 2 Shuijingtuo Formation of southern Shaanxi and western Hubei in South China. In this study, the shell architectures, epithelial cell moulds, and the shape and size of cylindrical columns are examined, shedding new light on our understanding of the architecture intricacy, biomineralization process, and evolutionary fitness of early phosphatic-shelled brachiopods.

# Results
## Systematic palaeontology

> Brachiopoda Duméril, 1806
> Linguliformea Williams, Carlson, Brunton, Holmer and Popov, 1996
> Lingulata Gorjansky and Popov, 1985
> Lingulida Waagen, 1885
> Linguloidea, Menke, 1828
> Eoobolidae, Holmer, Popov and Wrona, 1996

### Remarks

*Holmer et al., 1996* established the Eoobolidae to include lingulides characterized by a pitted metamorphic shell and a post-metamorphic shell with pustules. The new taxa described here are assigned to Eoobolidae based on these typical characters. Despite Balthasar's suggestion to reassign all Eoobolidae members to Zhanatellidae Koneva, 1986, based on the discovery of *Eoobolus* cf. *triparilis* from the Series 2 Mural Formation in the Canadian Rocky Mountains with a pitted metamorphic shell and tuberculate post-metamorphic shell (*Balthasar, 2009*), we adhere to Betts's argument for retaining Eoobolidae (*Betts et al., 2019*). Actually, the distinctive features of eoobolids, such as the elevated and divided ventral and dorsal pseudointerareas, are quite different from zhanatellids that are characterized by adpressed dorsal pseudointerarea (*Popov and Holmer, 1994*; *Betts et al., 2019*).

> Genus *Latusobolus* Zhang, Zhang and Holmer gen. nov.

### Type species

*Latusobolus xiaoyangbaensis* sp. nov., here designated.

### Etymology

From the Latin '*latus*' (wide), to indicate the transversely oval outline of both ventral and dorsal valves, morphologically similar to *Obolus*. The gender is masculine.

### Diagnosis

For a full description and discussion of *Latusobolus* gen. nov., refer to Appendix 1.

> *Latusobolus xiaoyangbaensis* Zhang, Zhang, and Holmer sp. nov.
> *Figure 1* and *Appendix 1—figures 1–4*, *Supplementary file 1*.

### Etymology

After the occurrence at the Xiaoyangba section in southern Shaanxi, China.

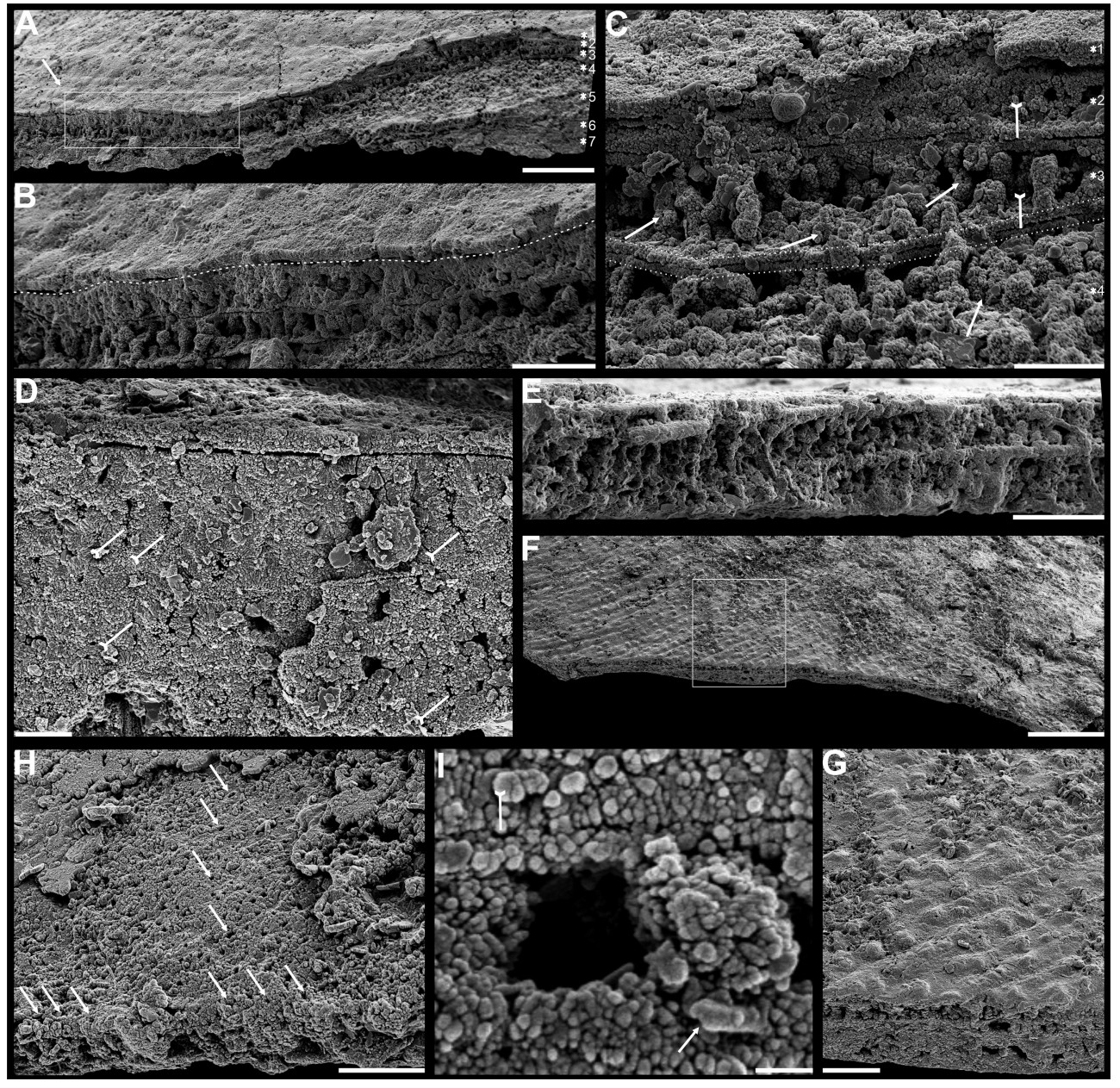

**Figure 1.** Shell architecture of *Latusobolus xiaoyangbaensis* gen. et sp. nov. from the Cambrian Series 2 Shuijingtuo Formation in southern Shaanxi, South China. (**A–C**) ELI-XYB S5-1 BS01. (**A**) Cross-section of a ventral lateral margin, note post-metamorphic pustules by arrow, primary layer 1 and stacked sandwich columnar units 2–7, box indicates area in **B**. (**B**) Enlarged view of (**A**) showing the boundary between top primary and underlying secondary layers by dotted line. (**C**) Enlargement of shell layers 1–4 of (**A**) note canals of columns (arrows), gap (tailed arrows) between two stratiform lamellae by dotted lines. (**D**) Poorly phosphatised columns of ventral valve, note canals by tailed arrows, ELI-XYB S5-1 BR06. (**E**) Columns of dorsal valve, ELI-XYB S5-1 BS17. (**F**) Cross-section of a ventral lateral margin, showing post-metamorphic pustules, box indicates the area in (**G**), ELI-XYB S4-2 BO06. (**G**) Enlarged primary layer pustules and underlying secondary layer columns. (**H**) Dorsal valve, one unit of stacked columnar architecture with the exfoliation of top primary layer, noting column canals on the stratiform lamella surface by arrows, ELI-XYB S4-2 BO08. (**I**) Apatite spherules of granule aggregations of ventral columnar shell structure, note granule rods by arrow and thin gap left by the degradation of organic counterparts by tailed arrow, ELI-XYB S4-2 BO06. Scale bars: (**A**), 50 µm; (**B**), (**E**), (**G**), 20 µm; (**C**), (**H**), 10 µm; (**D**), 5 µm; (**F**), 100 µm; (**I**), 1 µm.

## Type material

Holotype, ELI-XYB S5-1 BR09 (*Appendix 1—figure 1M–P*), ventral valve, and paratype, ELI-XYB S4-2 BO11 (*Appendix 1—figure 2M–P*), dorsal valve, from the Cambrian Series 2, level S5-1 and level S4-2, respectively, of Shuijingtuo Formation at the Xiaoyangba section (*Zhang et al., 2021a*) near Xiaoyang Village in Zhenba County, southern Shaanxi Province, China.

## Diagnosis

As for the genus.

## Description

For a full description and discussion of *Latusobolus xiaoyangbaensis* gen. et sp. nov., refer to Appendix 1.

Genus *Eoobolus Matthew, 1902*

## Type species

*Obolus* (*Eoobolus*) *triparilis Matthew, 1902* (selected by *Rowell, 1965*).

## Diagnosis

See Holmer et al. (p. 41) (*Holmer et al., 1996*).

*Eoobolus acutulus* Zhang, Zhang, and Holmer sp. nov.

*Figure 2* and *Appendix 1—figures 5–7*, *Supplementary file 2*.

## Etymology

From the Latin '*acutulus*' (somewhat pointed), to indicate the slightly acuminate ventral valves with an acute apical angle. The gender is masculine.

## Type material

Holotype, ELI-AJH S05 BT11 (*Appendix 1—figure 5E–H*), ventral valve, and paratype, ELI-AJH S05 1-5-07 (*Appendix 1—figure 5M*), dorsal valve, from the Cambrian Series 2, level S05 of Shuijingtuo Formation at the Aijiahe section (*Zhang et al., 2016b*) near Aijiahe Village in Zigui County, northwestern Hubei Province, China.

## Diagnosis

For a full description and discussion of *Eoobolus acutulus* sp. nov., refer to Appendix 1.

## Biomineralized columnar architecture

The shell architectures are exquisitely well-preserved in these newly assigned eoobolid *Latusobolus xiaoyangbaensis* gen. et sp. nov. and *Eoobolus acutulus* sp. nov. Their shell architectures are stratiform in a hierarchical pattern, and consist of laminated primary layer and columnar secondary layer (*Figures 1 and 2*). The laminated primary layer is about 1–3 µm thick, composed of compact apatitic lamellae (*Figures 1B and 2G*), while the secondary layer is stratiform with numerous columns disposed orthogonally between a pair of stratiform lamellae (*Figures 1B, C, 2B, C, J and K*; *Appendix 1—figure 4I and J*; *Appendix 1—figure 7J*) and looks like being composed of stacked sandwich columnar units. The hollow space in the columns and between lamellae of stacked columnar units may be originally filled with the rich composition of organic material (*Figures 1C, I and 2C, F, H* and *Figure 3E*; *Appendix 1—figure 4I*).

There are 1-3 layers of stacked sandwich columnar units developed in *Latusobolus xiaoyangbaensis* gen. et sp. nov. Columns are quite small about 2.4 µm in diameter, ranging from 1.6 µm to 3.4 µm, and about 6 µm in height, ranging from 2.9 µm to 11.9 µm. The central canal in the column ranges from 0.4 µm to 0.9 µm in diameter. The space between the stratiform lamellae is thin, around 0.7 µm, while the stratiform lamellae of columnar units are about 1.4 µm in thickness (*Figure 6—source data 1*).

The maximum number of multi-stacked sandwich columnar units increases to 13 in *Eoobolus acutulus* sp. nov. Columns are as small as in *Latusobolus xiaoyangbaensis* gen. et sp. nov., ranging

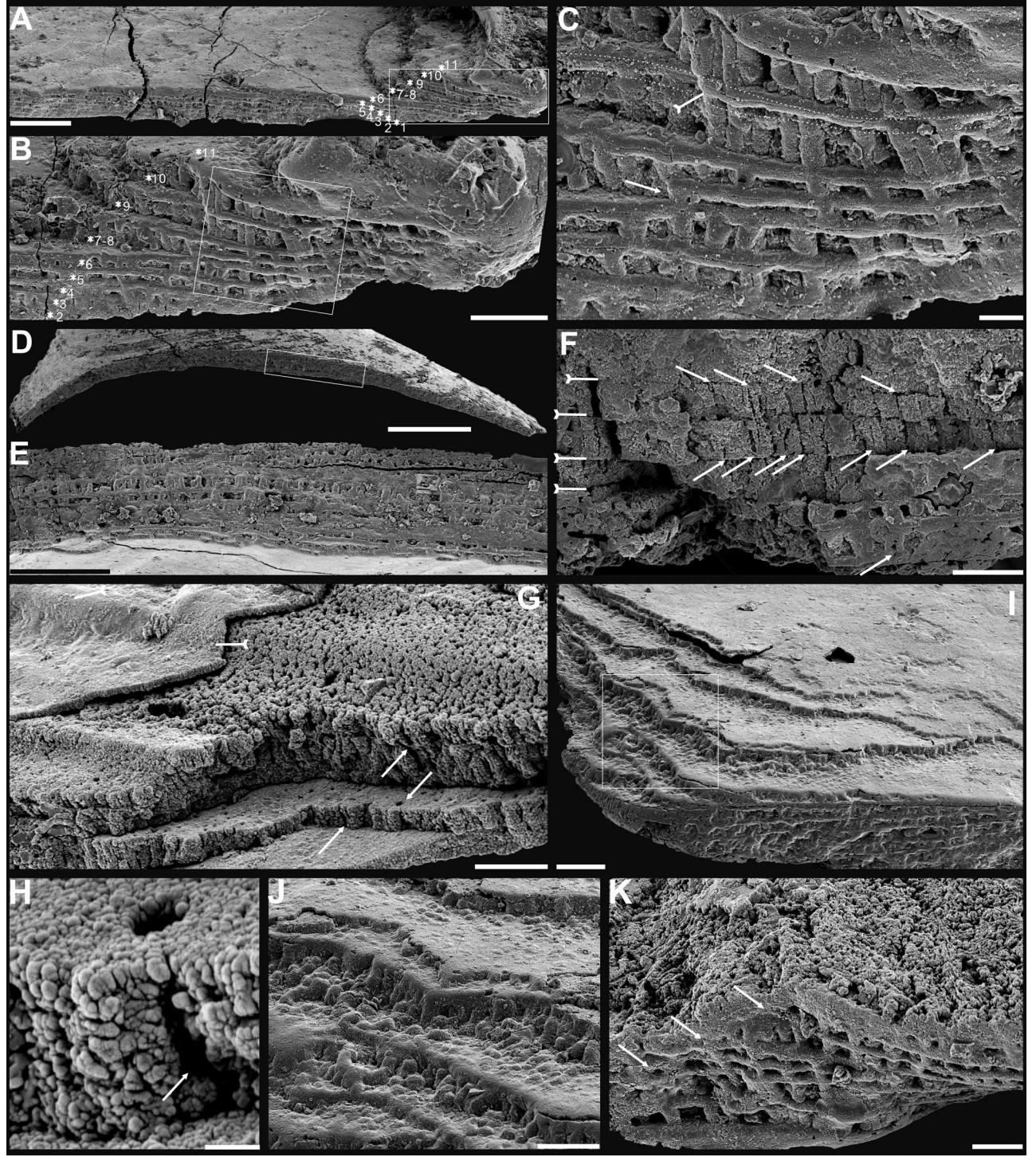

**Figure 2.** Shell architecture of ventral *Eoobolus acutulus* sp. nov. from the Cambrian Series 2 Shuijingtuo Formation in Three Gorges areas, South China. (**A–C**) ELI-AJH S05 BT12. (**A**) Cross-section of a ventral lateral margin, note primary layer 1 and stacked sandwich columnar units 2–11, box indicates area in **B**. (**B**) Enlarged view of **A**. (**C**) Enlarged view of **B**, show thin gap left by the degradation of organic counterparts (tailed arrow) between two stratiform lamellae by dotted lines, the fusion point of two columnar units by arrow. (**D–F**) ELI-AJH 8-2-3 BT02. (**D**) Cross-section of shell margin, box indicates area in **E**. (**E**) Different preservation condition of columnar architecture. (**F**) Poorly phosphatised columns, note the opening of canals along the organic membrane by arrows, and space between two stratiform lamellae by tailed arrows. (**G–H**) ELI-AJH 8-2-3 BT03. (**G**) Note canals on the cross-section and surface of stratiform lamella by arrows, and partly exfoliated primary layer by tailed arrow. (**H**) Magnified columns in (**G**), composed of granule spherules with canal by arrow. (**I**) Cross-section of shell margin, box indicates area in (**J**), ELI-AJH 8-2-3 BT04. (**J**) Enlarged short columns. (**K**) Imbricated columnar architecture (arrows), ELI-AJH S05 BT12. Scale bars: (**A**), (**E**), 50 µm; (**B**), (**I**), 20 µm; (**C**), 5 µm; (**D**), 200 µm; (**F**), (**G**), (**J**), (**K**), 10 µm; (**H**), 1 µm.

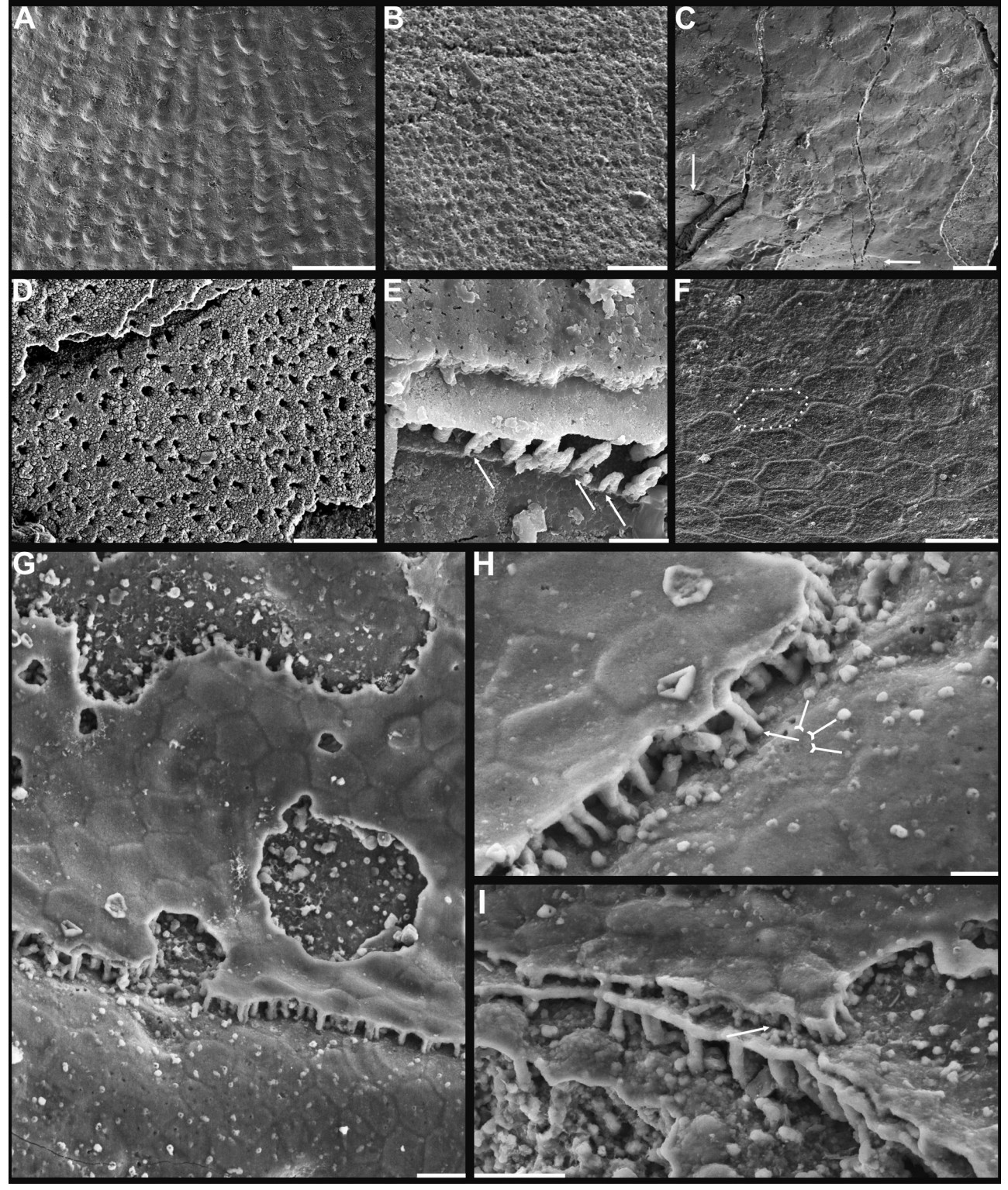

**Figure 3.** Shell ornamentation, ultrastructure, and epithelial cell moulds of Cambrian Series 2 brachiopods. (**A**) Post-metamorphic pustules of *Latusobolus xiaoyangbaensis* gen. et sp. nov., ELI-XYB S4-3 AU11. (**B–D**) *Eoobolus acutulus* sp. nov. (**B**) Metamorphic hemispherical pits, ELI-AJH 8-2-2 Lin01. (**C**) Epithelial cell moulds, note column openings on layer surfaces beneath by arrows, ELI-AJH S05 N31. (**D**) Enlarged column openings on layer surface, ELI-WJP 7 CE05. (**E–I**) *Eohadrotreta zhenbaensis*. (**E**) Partly broken columns, note canals by arrows, ELI-AJH F36. (**F**) Polygonal epithelial cell moulds on valve floor, dotted line indicates margin of one epithelial cell, ELI-WJP 6 R79. (**G–I**) ELI-AJH Acro 053. (**G**) Epithelial cell moulds on dorsal

*Figure 3 continued on next page*

*Figure 3 continued*

median septum with columns between them. (**H**) Enlarged view of **G**, note rudiment of columns by tailed arrows and one column on epithelial cell margin by arrow. (**I**) Epithelial cell moulds on stratiform lamella surfaces of successive three stacked sandwich columnar units developed on cardinal muscle areas with columns between (marked by arrow). Scale bars: (**A**), (**D**), 50 µm; (**B**), (**E**), (**F**), (**H**), 10 µm; (**C**), (**G**), (**I**), 20 µm.

from 1.2 µm to 3.2 µm, and about 4 µm in height. The central canal in the column is small, with the mean diameter of 0.7 µm. The space between stratiform lamellae of stacked columnar units is thin, around 0.6 µm, while the stratiform lamellae are about 1.2 µm in thickness (*Figure 6—source data 1*).

## Discussion
### Diversity of linguliform brachiopod shells

Although the supposed living fossil *Lingula* has long been considered to virtually lack morphological evolutionary changes (*Schopf, 1984*), more recent studies have shown that lingulide brachiopods have experienced dramatic modifications in many aspects (*Liang et al., 2023*), including arrangement of internal organs (*Zhang et al., 2008*), life mode (*Topper et al., 2015*), shell structure (*Cusack et al., 1999*), and even genome (*Goto et al., 2022*; *Luo et al., 2015*). The complexity and diversity of linguliform shell architecture was increasingly recognised in the pioneering study of Cusack, Williams and Holmer (*Cusack et al., 1999*; *Holmer, 1989*; *Williams and Cusack, 1999*; *Williams and Holmer, 1992*). Moreover, such complex architectures had a wide distribution in closely related brachiopod groups when they made their first appearance at the beginning of the Cambrian. In connection with an ongoing comprehensive scrutiny of well-preserved linguliform shell ultrastructures from the lower Cambrian limestones of South China, their complexity and diversity hidden in their conservative oval shape is becoming more and more intriguing. However, compared to their ancestral representatives, the shell structure in living lingulides is relatively simple, revealing profound modifications during their long evolutionary history (*Cusack et al., 1999*; *Holmer et al., 2008a*; *Williams, 1997a*; *Williams and Cusack, 1999*).

In general, the shells of organo-phosphatic brachiopods are stratiform, composed of an outer periostracum and inner rhythmically-disposed succession of biomineralized lamellae or laminae (*Holmer, 1989*; *Williams et al., 1997b*). The organic periostracum, serving as a rheological coat to the underlying shell, is rarely fossilized. However, its wrinkling and vesicular features have largely been preserved as superficial imprints (pits, pustules, fila, grooves, ridges, rugellae, drapes, reticulate networks, and spines) on the surface of the primary layer (*Figure 3A and B*), and are important characters in understanding brachiopod phylogeny (*Holmer, 1989*; *Holmer et al., 1996*; *Cusack et al., 1999*). By contrast, the structures preserved in the secondary layer have been characterised in fossil and living groups by three ancient fabrics –columnar, baculate, laminated – all of which persist in living shells except for the columnar fabric (*Cusack et al., 1999*). The tertiary shell layer is well developed in some recent and Palaeozoic lingulides (*Holmer, 1989*), but it is not recognised in the early eoobolides.

The primary layer commonly consists of heavily biomineralized compact laminae composed of apatite granules, with a thickness from 2 µm to 20 µm (*Williams, 1997a*; *Figures 1B, C and 2G*; *Appendix 1—figure 4J*). Usually, the concentric growth lines are evenly distributed on the surface of the primary layer of the post-metamorphic shell. By contrast, the surface ornamentation tends to be unevenly distributed, demonstrating a strong phylogenetic differentiation. Superficial pustules are one of the most distinct patterns and are readily recognised in one of the oldest brachiopod groups, the Eoobolidae (*Holmer et al., 1996*). The pustules are roughly circular in outline, composed of apatite aggregates, and range from 2 µm to 20 µm in diameter (*Zhang, 2018*; *Figures 1G and 3A*; *Appendix 1—figure 6J*). Such pustules are also found on early obolid, zhanatellid, and acrotheloid shells with relatively wide size variations from 5 µm to 30 µm in diameter (*Cusack et al., 1999*; *Zhang, 2018*). Although different in size, the similar pattern may indicate the same secretion regime that originated as vesicles during the very early stages of periostracum secretion (*Cusack et al., 1999*). The thickness of the underlying secondary layer varies greatly in different brachiopod groups, depending on the shell component and fabric type. Columnar, baculate, and laminated fabrics are incorporated into the basic lamination component to form the diverse stratiform successions of the secondary shell layer.

The fossil record reveals that the columnar shell structure (*Figure 3G–I* and *Figure 4*) is generally preserved in most early linguliforms (*Holmer et al., 2008b*; *Streng et al., 2007*; *Williams et al., 1997b*; *Zhang, 2018*; *Zhang et al., 2021a*). It is a multi-stacked sandwich architecture (multi-columnar units are stacked in a vertical direction), which is developed in the earliest linguliform *Eoobolus* with a relatively simple architecture as an early developmental stage (*Zhang et al., 2021a*). Each stacked sandwich columnar unit consists of numerous columns disposed orthogonally between a pair of compact stratiform lamellae (*Cusack et al., 1999*; *Holmer, 1989*). One to three stacked sandwich columnar units with short orthogonal columns can also be found in *Latusobolus xiaoyangbaensis* gen. et sp. nov. (*Figure 1*) and *E. incipiens* (*Figure 4A*), while the number of stacked sandwich units increased in later eoobolids (*Figure 2*), obolids and lingulellotretids (*Figure 4B–D*). Eventually, the acrotretides developed a more complex columnar architecture with multiple stacked sandwich units (*Figure 4F–J*). Moreover, columnar shell structures are also found in stem group brachiopods, e.g., *Mickwitzia*, *Setatella*, and *Micrina*, but with different column size and numbers of laminae (*Butler et al., 2015*; *Holmer et al., 2002*; *Holmer et al., 2008a*; *Skovsted et al., 2010*; *Williams and Holmer, 2002*). It is assumed here that the columnar architecture may be a plesiomorphic character in linguliform brachiopods, inherited from stem group brachiopods.

## Biomineralization process of organo-phosphatic columnar architecture

Metazoans are known for secreting very different types of biominerals through the process of biological mineralization. This linking of living soft organic tissues with solid earth minerals is a process that has changed the nature of Earth's fossil archive (*Addadi and Weiner, 2014*; *Lowenstam and Weiner, 1989*; *Roda and Mar, 2021*; *Wood and Zhuravlev, 2012*). Because of the fine quality of phosphate biomineralization in linguliforms (*Cusack et al., 1999*; *Williams and Cusack, 1999*), they can have exquisitely finely preserved shell ultrastructures (*Figures 3 and 4*), including epithelium cell moulds (*Figure 3C and F–I*). This permits us to reconstruct the biomineralization process of their apatitic cylindrical columns and address key questions about how these hierarchical structures relate to mechanical functions. Although the biomineralization process of living brachiopods at the cellular level is not well known, biochemical experiments (*Cusack et al., 1999*; *Cusack et al., 1992*; *Lévêque et al., 2004*; *Williams and Cusack, 1999*) have revealed the possibility that the biologically-controlled, organic matrix mediated extracellular mineralization during brachiopod shell secretion. This process can be compared to the hard tissue-forming process of mollusc shells and vertebrate teeth (*Golub, 2011*; *Neary et al., 2011*; *Roda and Mar, 2021*).

Many polygonal structures (*Figure 3C and F*), preserved on the internal surface of successively alternating laminae (*Figure 3G–I*), have generally been considered to represent the moulds of epithelial cells (*McClean, 1988*). The average size of epithelial cells is 20 μm, ranging from 5 to 30 μm in early Cambrian linguliforms (*Zhang et al., 2016b*). The rheological vesicle environment may cause the shape variation of epithelia, while different secreting rates could result in different sizes. Generally, smaller epithelial cells had higher secretion rates (*Zhang et al., 2016b*). As the active secretion activity of outer epithelial cells, they can be easily embedded in the newly formed bounding surfaces of shell laminae (*McClean, 1988*; *Winrow and Sutton, 2012*), which left shallow grooves between epithelial margins as intercellular boundaries (*Figure 3F*). Epithelial cells are preserved as moulds – presumably by the phosphatization of smooth organic sheets (*Figure 4G and H*) – that had been secreted relatively slowly by the outer plasmalemma of the outer mantle (*Cusack et al., 1999*).

These active cells are responsible for the secretion of the linguliform periostracum with rheomorphic wrinkling features (*Cusack et al., 1999*), which left typical microtopography on the external surface of the underlying primary layer (*Figure 3A and B*). On the other hand, they also secreted linguliform biominerals in a rheological extracellular environment. It is inferred from living *Lingula*, that apatite grows from an amorphous calcium phosphate precursor, which forms the basic crystals of apatite around 5 nm in diameter (*Lévêque et al., 2004*; *Williams and Cusack, 1999*). These nanoscale crystals were packaged into apatite granules with an average size of 100 nm (*Figures 1I and 2H*; *Appendix 1—figure 4K*), which acted as the fundamental component building up the hierarchical stratiform shells, including the primary layer and secondary layer (*Figure 5*). The apatite granules were probably coated or saturated with organic compounds to form granule aggregations or clusters (spherular mosaics) as irregular spherules or rods of about 500 nm (*Figures 1I and 2H*; *Appendix 1—figure 4J*, *Appendix 1—figure 7J and K*), usually leaving gaps between the aggregation boundaries

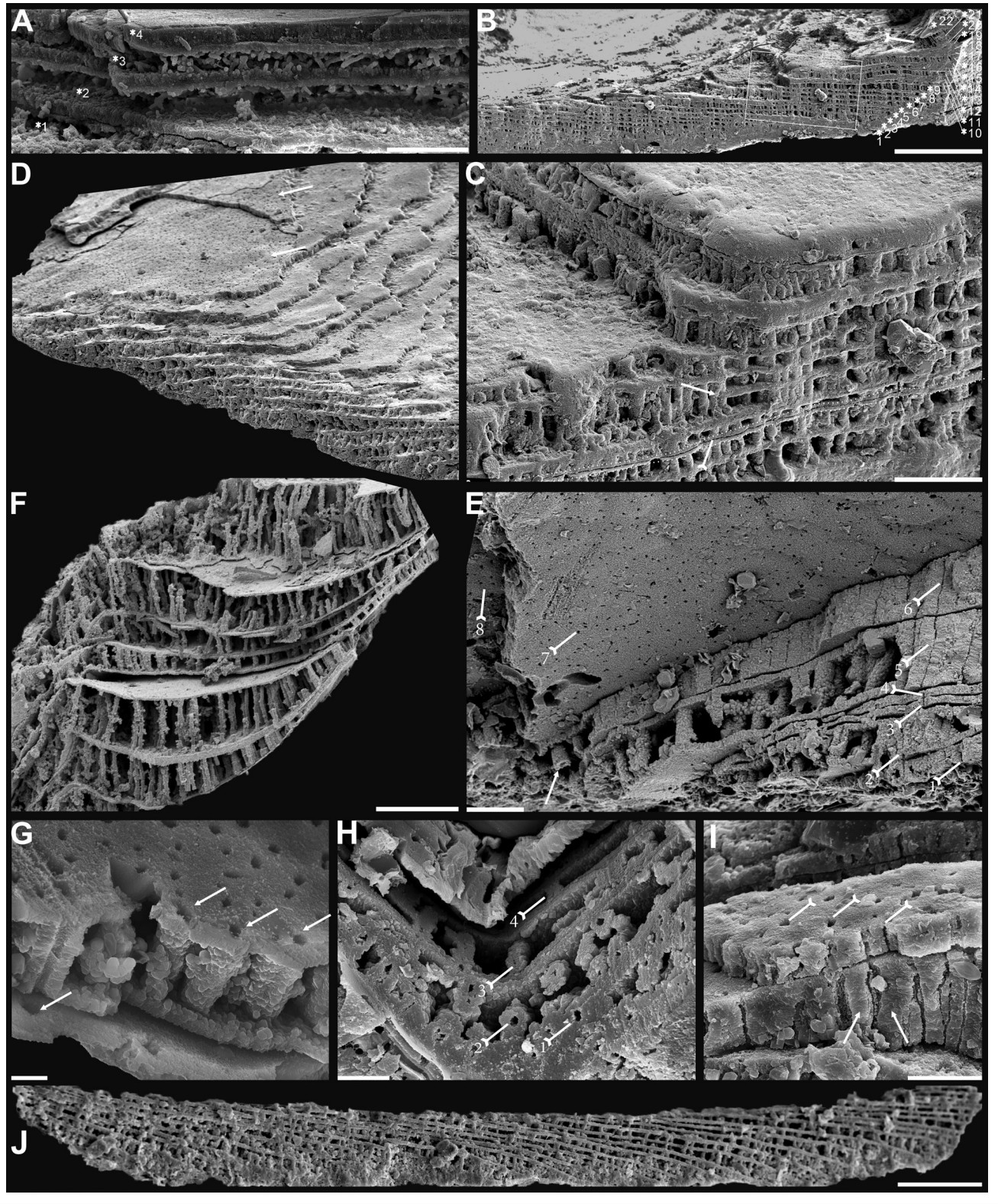

**Figure 4.** Biomineralized columnar architecture of Cambrian Series 2 brachiopods. (**A**) *Eoobolus incipiens*, P00712-AJXM-267.5 DT-12. (**B–D**) *Lingulellotreta ergalievi*, ELI-AJH 8-2-3 CI11. (**B**) Cross-section of shell margin, box indicates area in **C**, note primary layer 1 and stacked sandwich columnar units 2–22, and raised pseudointerarea (tailed arrow). (**C**) Enlarged view of thin gap (tailed arrow) between two stratiform lamellae by dotted lines, the fusion of two stacked columnar units into one by arrow. (**D**) Imbricated growth pattern of stacked columnar units. (**E**) *Palaeotreta zhujiahensis*,

*Figure 4 continued on next page*

*Figure 4 continued*

note column openings (arrow) on eight successive columnar units by tailed arrows, ELI-AJH 8-2-1 AE09. (**F–J**) *Eohadrotreta zhenbaensis*. (**F**) Relatively taller columns (ca. 20 μm), ELI-AJH 8-2-1 acro16. (**G**) Apatite spherules of granule aggregations in one columnar unit, note column openings (arrows) on both stratiform lamella surfaces, ELI-AJH S05 E18. (**H**) Cross-section shows column openings on four successive units by tailed arrows, ELI-WJP 7 AB98. (**I**) Poorly phosphatised columns (arrows), note openings of canals on surface of stratiform lamella by tailed arrows, ELI-AJH S05 I76. (**J**) Stacked columnar units in an imbricated pattern, ELI-WJP 6 R47. Scale bars: (**A**), (**E**), (**I**), 10 μm; (**B**), 100 μm; (**C**), (**F**), 20 μm; (**G**), 2 μm; (**H**), 5 μm; (**D**), (**J**), 50 μm.

in fossils after the degradation of organic counterparts (*Figure 2H*; *Appendix 1—figure 4K*). These apatite spherules are aggregated in the planar orientation as compact thin lamella less than 4 μm in thickness. Several thin lamellae are closely compacted to form the primary layer (*Figure 1B and C*; *Appendix 1—figure 4I and J*). On the other hand, similar nanometre scale networks of spherules are aggregated and organised as orthogonal columns perpendicular to a pair of stratiform lamella surfaces, forming one columnar unit. Multi-columnar units are stacked in a vertical direction from the exterior to the interior to form the secondary layer, applying a stacked sandwich model (*Figure 5*), which differs from the layer cake model. A very thin gap, commonly less than 1 μm, between each pair of stacked sandwich columnar units is obvious in well-preserved specimens (*Figures 1C, I, 2C, F, and 4C*; *Appendix 1—figure 4I*; *Appendix 1—figure 7J*). This is likely indicating an organic membrane acting as an extracellular matrix with functions of a template guiding mineral nucleation (*Addadi and Weiner, 2014*; *Cusack et al., 1999*; *Lévêque et al., 2004*). This is also supported by the modular nature of the columnar architecture, revealing a homogeneous organic substrate responsible for the succeeding rhythmic sequence. Although, newly secreted columnar units may succeed the older one unconformably with overlap (*Figures 2C, 3I and 4D*) and be involved in lateral changes of column size (*Figures 2K and 4C*), it is supposed to reflect intracellular deviations with the same secretory cycle of the outer mantle as a whole as in living lingulids (*Cusack et al., 1999*; *Williams et al., 1992*). The secreting and building processes of the early Cambrian phosphatic-shelled brachiopod columnar shells, secreted by the underlying outer epithelium cells of the mantle lobe, are illustrated in *Figure 5*.

It is worth noting that, on well-preserved specimens nanoscale openings are permeated on the terminal ends of the orthogonal columns (*Figures 1C, 4E and H*) and surfaces of stratiform lamellae

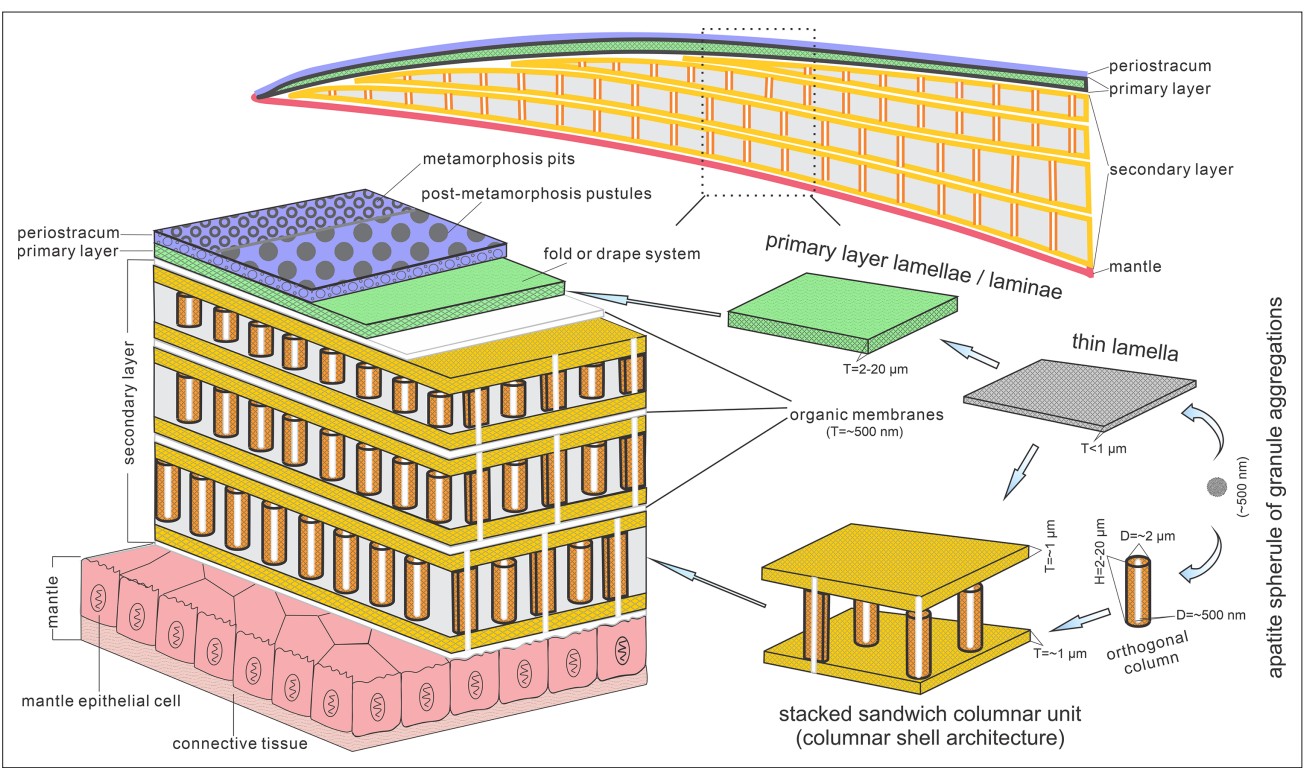

**Figure 5.** Biomineralization process of typical columnar architecture using the stacked sandwich model of phosphatic-shelled brachiopods. Abbreviation: D=Diameter; H=height; T=Thickness. modified from *Williams and Holmer, 1992* (Text-Fig. 7); *Zhang et al., 2016b* (Figure 6).

(*Figures 1H, 2G, 3D, E, 4G and I*). The openings are rounded with a mean diameter of 600 nm. Observation through natural fractures of shells shows that some canals can be traced continuously through several columnar sequences (*Figures 1D, 2F and 4E*), while in poorly preserved fossils, they are filled with secondarily phosphatised spherules or mosaics (*Figures 1I and 2G*), occasionally leaving random gaps (*Figures 1D, 2F and 4I*). These canals are very likely the voids left by degraded organic material, which is confirmed by different taphonomic processes that preserve canals in contemporaneous early Cambrian Burgess Shale–type fossil Lagerstätte in South China (*Duan et al., 2021*). The regular arrangement of the canal systems and closely related columns reveals that the process is organic matrix-mediated. Furthermore, the even disposition of the organic matrix in columns indicates the existence of rheological and central areas, on which biologically controlled biomineralization took place. It reveals that the nucleation, growth, and aggregation of the deposited amorphous calcium phosphate are directed by the same group of epithelial cells (*Cusack et al., 1999*; *Pérez-Huerta et al., 2018*; *Roda and Mar, 2021*; *Weiner and Dove, 2003*). In the stacked sandwich columnar units, the empty chambers between each column (*Figure 4F and J*) would be originally filled with glycosaminoglycans (GAGs) as in living *Lingula* (*Cusack et al., 1999*; *Williams et al., 1994*). These chambers are often filled with coarse spherular mosaics when being secondarily phosphatised and consequently, they become indistinguishable from the columns, paired stratiform lamellae and organic membrane (*Figures 1D, 2E and 4I*). At the posterior margin of mature shells, especially the ventral pseudointerarea where the vertical component of the growth vector becomes increasingly important in ventral valves, the short columns are succeeded by relatively taller columns, resulting in the change from two stacked sandwich columnar units fused into one unit during growth anteriorly (*Figures 2C K, 3I and 4D*). This may demonstrate allometric growth of the shell (*Cusack et al., 1999*).

The most intriguing and enigmatic phenomenon of skeletal biomineralization is the evolutionary selection of calcium carbonate and calcium phosphate in invertebrates and vertebrates, respectively (*Lévêque et al., 2004*; *Luo et al., 2015*). However, the Brachiopoda is a unique phylum that utilises both minerals. The appearance of apatite as a shell biomineral dates back as far as Cambrian Age 2 for stem group brachiopods (*Skovsted et al., 2015*; *Topper et al., 2013*; *Ushatinskaya, 2002*) and persists to the present in living linguliforms (*Carlson, 2016*). Calcium phosphate can build a relatively less soluble skeletal component compared with calcium carbonate shells, but with the disadvantage of a greater energetic and physiologic cost (*Wood and Zhuravlev, 2012*). The acquisition of this specific biomineral in phosphatic-shelled brachiopods has been considered an ecological consequence of the globally elevated phosphorous levels during phosphogenic event in the calcite seas with a low Mg:Ca ratio and/or high $CO_2$ pressure (*Balthasar, 2009*; *Brasier, 1990*; *Cook and Shergold, 1984*; *Wood and Zhuravlev, 2012*). In such situations, linguliforms were able to utilise sufficient phosphorous in ambient waters, unlike brachiopod ancestors that possessed an unmineralized shell coated with detrital grains, like *Yuganotheca* found in the Chengjiang Lagerstätte (*Zhang et al., 2014*). The acquisition of a calcium phosphatic shell may have been an evolutionary response of prey to an escalation of predation pressure during the Cambrian explosion of metazoans (*Cook and Shergold, 1984*; *Wood and Zhuravlev, 2012*). Consequently, organic-rich biomineral composites of linguliform brachiopod shells possessed innovative mechanical functions, providing competitive superiority and adaptation on Cambrian soft substrates as well as reducing susceptibility to predation (*Wood and Zhuravlev, 2012*). Despite the physiological cost of calcium phosphate biomineralization and subsequent reduction in phosphorus levels during post Cambrian period, linguliforms have retained their phosphate shells during dramatical oscillations of seawater chemistry and temperature over 520 million years.

Given the long history of this subphylum, the possession of a phosphatic shell likely has numerous advantages. The innovative columnar architecture can mechanically increase the thickness and strength of the shell by the presence of numerous, stacked thinner laminae, comparable with the laminated fabric seen in obolids (*Cusack et al., 1999*; *Zhang et al., 2016a*). Furthermore, the stacked sandwich columns also increase the strength, flexibility, and ability to resist crack propagation by filling the space between the stratiform lamellae with organic material, comparable with the baculate fabric (*Lévêque et al., 2004*; *Merkel et al., 2009*). Thus, the stacked sandwich model of the columnar architecture possesses a greater advantage of mechanical functions and adaptation with a superior combination of strength, durability, and flexibility in laminated and baculate fabrics, resembling the colonnaded and reinforced concrete often used in urban construction. New data from nuclear magnetic resonance spectroscopy and X-ray diffraction reveals that apatite in brachiopod shells is

highly ordered and thermodynamically stable crystalline and it is more robust in the extremes of moisture, ambient osmotic potential and temperature, unlike the poorly ordered crystal of vertebrate bone (*Neary et al., 2011*). This type of more efficient and economical shell may also have been responsible for the early diversity of major linguliform brachiopods during the Cambrian explosion, resulting in this group becoming a significant component of the Cambrian Evolutionary Fauna (*Bassett et al., 1999*; *Sepkoski, 1984*; *Zhang et al., 2008*; *Zhang et al., 2020b*; *Zhang et al., 2021b*).

## Evolution of stacked sandwich columnar architecture in early brachiopod clades

Evolutionary transformations have repeatedly modified the organo-phosphatic architecture consisting of various aggregates of spherular apatite, held together by a scaffolding of glycosaminoglycan complexes, fibrous proteinaceous struts and chitinous platforms, in linguliform brachiopod shells since the early Cambrian (*Cusack et al., 1999*). As one of the oldest forms of brachiopod shell architectures, the columnar shell has long been regarded as a unique character of acrotretide brachiopods (*Cusack et al., 1999*; *Holmer, 1989*). However, recent discoveries of columnar shell structures in a diversity of early Cambrian stem group brachiopods have revealed that the same biomineralization strategy is utilised much more widely than previously thought (*Butler et al., 2015*; *Holmer et al., 1996*; *Holmer et al., 2008a*; *Skovsted et al., 2010*; *Streng et al., 2007*; *Ushatinskaya and Korovnikov, 2014*; *Zhang, 2018*; *Zhang et al., 2021a*). This highlights the need for a better understanding of the origin and adaptive modification of stacked-sandwich columnar architectures in early lophophorate evolution.

The Eoobolidae is presently considered to be the oldest known linguliform brachiopods (*Zhang et al., 2021a*). The biomineralized orthogonal columns in *Eoobolus incipiens* from lower Cambrian Stage 3 probably represent an early and simple shell structure type with a poorly developed columnar secondary layer (*Figure 4A*). The columns are relatively small with a mean diameter of 1.8 µm ranging from 0.6 to 3.0 µm, and a mean height of 4.1 µm (*Figure 6*; *Figure 6—source data 1*). The complete secondary layer is only composed of two to three stacked sandwich columnar units, resulting in a shell thickness of about 30 µm. Such a simple shell structure is also developed in the slightly younger *Latusobolus xiaoyangbaensis* gen. et sp. nov. (*Figure 1*), but with a slightly taller column of about 6.2 µm. From Cambrian Age 4, the number of stacked sandwich columnar units increases rapidly in the Eoobolidae, growing to as many as 10 stacked sandwich columnar units in *Eoobolus acutulus* sp. nov. (*Figure 2*), and as many as 10 in *Eoobolus*? aff. *priscus* with the shell thickness of about 50 µm (*Streng et al., 2007*). However, the size of individual columns keeps within a stable range, around 4 µm in height and 2 µm in diameter. Compared with Eoobolidae, the Lingulellotretidae demonstrates a more developed columnar shell, which has a relatively larger number of stacked sandwich columnar units (up to 20), effectively increasing the shell thickness to 70 µm. The column size is very similar in both *Lingulellotreta malongensis* and *L. ergalievi* (*Figure 4B and D*), and matches that of the contemporaneous *Eoobolus*. But the slightly younger *L. ergalievi* has more columnar units, resulting in a thicker shell than *L. malongensis*.

Among all early Cambrian linguliforms with columnar architectures, the acrotretides have developed the most complex shell structure (*Figure 4E–J*). The Cambrian fossil record unveiled a clear pattern of increasing growth (regarding both the diameter and height of the columns and the number of stacked sandwich columnar units) of the columnar architecture in acrotretides: from a very simple type, observed in *Palaeotreta shannanensis* (similar to that of *E. incipiens*) to the slightly more developed structure in *Palaeotreta zhujiahensis* (similar to that of *L. malongensis*) (*Zhang et al., 2020c*) to the most advanced architecture observed in *Eohadrotreta zhenbaensis* and younger specimens (*Figure 6*). The diameter of a single orthogonal column increases about two times in acrotretides compared to eoobolids, whereas the general height of the columns increases to 10 µm in *Eohadrotreta zhenbaensis* and to 29 µm in *Hadrotreta primaeva*, which is about 10 times as high as seen in *Eoobolus variabilis*. Furthermore, the number of columnar units has also increased to about 30, collectively increasing the shell thickness to a maximum value of more than 300 µm in *Eohadrotreta*.

The homology of the columnar architecture in early linguliforms outlines a clear picture, most likely representing a continuous transformation between the Lingulida (Eoobolidae and Lingulellotretidae) and Acrotretida. Two groups that represent a major component of early Cambrian benthic communities (*Chen et al., 2021*; *Claybourn et al., 2020*; *Topper et al., 2015*; *Zhang et al., 2008*; *Zhang,*

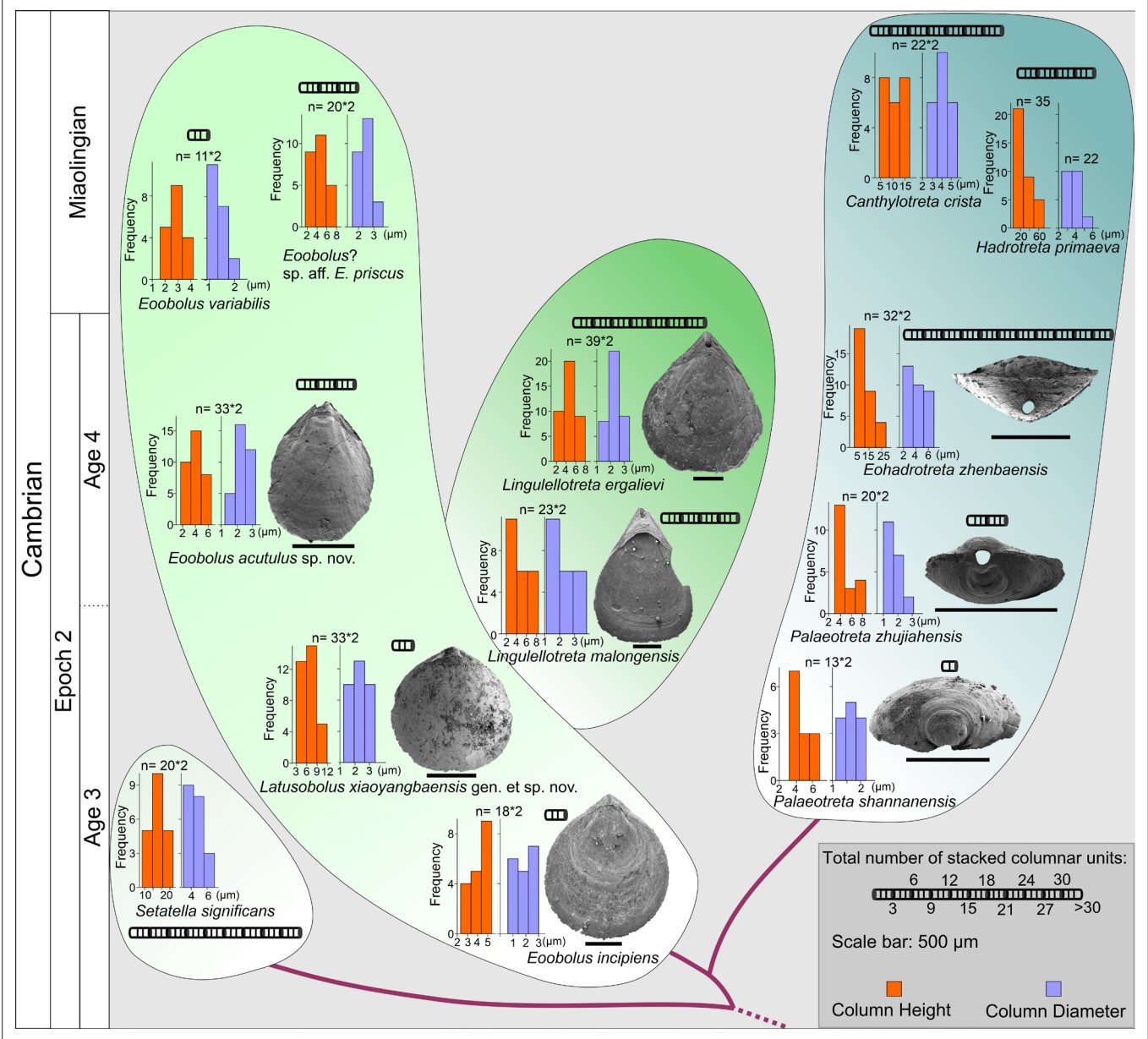

**Figure 6.** The evolution of stacked sandwich columnar architecture in early Eoobolidae taxa, *Eoobolus incipiens*, *Latusobolus xiaoyangbaensis* gen. et sp. nov., *Eoobolus acutulus* sp. nov., *Eoobolus variabilis*, *Eoobolus*? aff. *priscus*, Lingulellotretidae *Lingulellotreta malongensis*, *Lingulellotreta ergelievi*, Acrotretida *Palaeotreta shannanensis*, *Palaeotreta zhujiahensis*, *Eoohadrotreta zhenbaensis*, *Hadrotreta primaeva*, *Canthylotreta crista,* and stem group *Setatella significans*. The height and diameter data of columns are based on data from literature (*Skovsted and Holmer, 2003*; *Streng et al., 2007*; *Streng and Holmer, 2006*; *Ushatinskaya and Korovnikov, 2014*; *Zhang et al., 2016b*; *Zhang et al., 2020b*; *Zhang et al., 2020c*).

The online version of this article includes the following source data for figure 6:

**Source data 1.** Raw data of the measurements of diameter and height of columns, thickness of different shell layers and number of columnar units demonstrated in *Figure 6*.

*2018*; *Zhang et al., 2020b*). The possible occurrence of this shell architecture within the family Obolidae cannot be discounted, as detailed information on the possible columnar shell structures in early Cambrian representatives such as *Kyrshabaktella* and *Experilingula* are poorly known (*Cusack et al., 1999*; *Streng et al., 2007*; *Williams, 1997a*). The small size and simple pattern of stacked sandwich columnar architectures remain stable in the Eoobolidae, and this stability likely limits both the shell thickness and overall body size, with ventral and dorsal valves of the family remaining below millimetre until Miaolingian (middle Cambrian). Stacked sandwich columnar architectures are a character

state of the *Lingulellotreta* shell structure as well; however more columnar units are developed that slightly increase the shell thickness and subsequently species of *Lingulellotreta* reach body size about twice that of *Eoobolus* (*Zhang et al., 2004*). A continuous transformation of anatomic features can be deduced from the evolutionary growth of columnar shells between the two clades. First, the orthogonal columns are markedly developed at the pseudointerarea area of the ventral valve of *Lingulellotreta* (*Figure 4B*), resulting in a greater elevation of the pseudointerarea above the shell floor. It leaves a large amount of space for the posterior extension of the digestive system, which is well protected by the covering mineralized ventral pseudointerarea. This is supported by the discovery of a curved gut under the pseudointerarea of *Lingulellotreta malongensis* in the Chengjiang Lagerstätte (*Zhang et al., 2007*). Second, with the continuous growth of the ventral pseudointerarea, the opening obolide-like pedicle groove is sealed, resulting in a unique pedicle opening exclusively observed in the lingulellotretid brachiopods. Thus, the early pedicle-protruding foramen between the ventral and dorsal valves of the Linguloidea is transformed into a new body plan where the pedicle opening is restricted to the ventral valve of lingulellotretids. It supports the scenario that the columnar architecture is monophyletic in at least Linguloidea, and that the slightly younger lingulellotretid columnar architecture was derived from an *Eoobolus*-like ancestor during late Cambrian Age 3.

In another evolutionary direction, acrotretide brachiopods fully utilise the columnar architecture including the derived camerate fabric as shell structure across the whole clade (*Streng and Holmer, 2006*). The similarity and gradually evolutionary transformations of shell structure from simple forms in lingulides to complex forms in acrotretides suggests the stacked sandwich columnar architecture did not evolve independently in actrotretides. In terms of the derived camerate fabric, more mineralized material is utilised compared to its precursor, the columnar shell structure (*Streng et al., 2007*). The column size, including height and the number of stacked sandwich columnar units uniformly increase to about 10 times greater in acrotretides than in *E. incipiens* and *L. xiaoyangbaensis*, since late Cambrian Age 3 (*Figure 6*). Despite the increase in size of the columns and the number of stacked sandwich columnar units, the whole body of acrotretides is restricted to only millimetre size (*Holmer, 1989*; *Popov and Holmer, 1994*). A continuous transformation of anatomic features and shell structure functions can be deduced from the evolutionary growth of columnar shells in early acrotretides. Firstly, the stacked sandwich columns were markedly developed at the posterior area of the ventral valve, resulting in a greater elevation of the pseudointerarea above the valve floor, compared to that of *Lingulellotreta*. Second, two transformations subsequently changed an obolid-like ventral valve (*Palaeotreta shannanensis*) to a cap shape valve (*P. zhujiahensis*) (*Zhang et al., 2020c*), and a conical shape (*Eohadrotreta zhenbaensis*) (*Zhang et al., 2018*), and eventually to a tubular shape (*Acrotreta*) (*Holmer and Popov, 1994*). During this evolution, dorsal valves remained relatively flat, showing a limited height profile. The obolid-like ventral pseudointerarea changed from orthocline to catacline and eventually to procline with strongly reduced propareas, while the ventral muscular system moved towards the elevated posterior floor, resulting in the formation of a new apical process (*Popov, 1992*; *Zhang et al., 2018*). With the increasing growth of the stacked sandwich columnar shells, the thick organo-phosphatic shell may have increased in strength providing more mechanical support to the conical or tubular valve in a turbulent environment.

Based on the similar evolutionary trajectory, regarding the increasing growth of the shell in Lingulellotretidae and Acrotretida, the formation of their ventral pedicle foramens is very likely homologous, modified from an obolid-like pedicle groove between the two valves. Furthermore, the similarity and continuity in the increasing number and size of the orthogonal columns suggest that columnar architecture is a plesiomorphic character in Linguloidea and Acrotretida. However, the phylogenetic puzzle of whether the columnar architecture is paraphyletic with the baculate fabric in Linguliformea, or even in Lingulata hangs on two pieces of important fossil evidence. The shell structure of the earliest linguliform brachiopods on a global scale needs to be comprehensively investigated based on better preserved fossils. Moreover, more extensive scrutinization of the shell architecture and composition in widely distributed early Cambrian stem group brachiopods is required to conclusively resolve their phylogenetic relationship with linguliforms. The complex shell in stem group taxa *Setatella* and *Mickwitzia*, that are younger and have more advanced columnar shell features than *Eoobolus incipiens*, might reveal the plesiomorphic state of the columnar architecture in Linguliformea (*Butler et al., 2015*; *Holmer et al., 2008a*; *Skovsted and Holmer, 2003*; *Williams and Holmer, 2002*). The accuracy of this assumption depends on future work and whether the columnar shell structure is preserved

in older ancestors and other stem group taxa. In another scenario, baculate, laminated, and columnar architectures might have originated independently from an unmineralized ancestor like the agglutinated *Yuganotheca* during the early Cambrian (*Cusack et al., 1999*; *Zhang et al., 2014*). Regardless of what scenario is true, the origin of the innovative columnar architecture with a stacked sandwich model has played a significant role in the evolution of linguliform brachiopods. The evolutionary diversity of shell architectures would match the general increase in the diversity of phosphatic-shelled brachiopods during the Cambrian radiation. Among them, the micromorphic acrotretides demonstrate the superb application of the columnar architecture combined with its innovative conical shape and possible exploitation of secondary tiering niches (*Topper et al., 2015*; *Wang et al., 2012*; *Zhang et al., 2018*). The fitness of the diminutive body size of acrotretides is likely a trade-off between the increasing metabolic demand of phosphate biomineralization after the Cambrian phosphogenic event and the increased chance of evolutionary survival and adaptation by producing a high mechanical skeleton for protection in the shallow water environment (*Cook and Shergold, 1984*; *Garbelli et al., 2017*; *Lévêque et al., 2004*; *Neary et al., 2011*; *Roda and Mar, 2021*; *Wood and Zhuravlev, 2012*). Their relatively large surface/volume ratio mechanically requires strong support from the composition of stacked sandwich columnar architecture and possibly a relatively lower density of the shell by organic biomineralized material for the secondary tiering life. Such adaptive innovations may account for the flourish of phosphatic-shelled acrotretides in the latter half of the Cambrian, continuing to the Great Ordovician Biodiversification Event, thriving and playing an important role in marine benthic communities for more than 100 million years.

## Materials and methods

The brachiopod material studied here was collected from the Cambrian Series 2 Shuijingtuo Formation at the Xiaoyangba section of southern Shaanxi (*Zhang et al., 2021a*), and the Shuijingtuo Formation at the Aijiahe section and Wangjiaping section of western Hubei (*Zhang et al., 2016b*). All specimens are recovered through maceration of limestones by acetic acid (~10%) in the laboratory, and deposited in the Early Life Institute (ELI), Northwest University, China. Selected specimens were coated and studied further using Fei Quanta 400-FEG SEM at Northwest University, Zeiss Supra 35 VP field emission at Uppsala University, and JEOL JSM 7100F-FESEM at Macquarie University. Measurements of length, width, and angle of different parts of *Latusobolus xiaoyangbaensis* gen. et sp. nov. and *Eoobolus acutulus* sp. nov. are performed on SEM images of well-preserved specimens by TpsDig2 v. 2.16. Measurements of diameter and height of orthogonal columns and thickness of different shell layers were performed on SEM images of available adult specimens from this study and previously published literatures by TpsDig2 v. 2.16. Shell thickness was measured at the posterior region of both ventral and dorsal valves of available adult specimens, where the shell displays maximum thickness. The number of columnar units was also counted in the posterior region of available adult specimens. Raw data is provided in *Supplementary file 1*, *Supplementary file 2* and *Figure 6—source data 1*.

## Acknowledgements

We would like to thank Profs. LE Popov, GA Brock, Y Cai, and CY Cai for insightful discussion, and QC Feng, JP Zhai and CM Han for sample preparation. Thanks to S Lindsay and C Shen at Microscopy Unit at Macquarie University, M Streng at Uppsala University, and YL Pang at Northwest University for assistance with SEM imaging. Thanks also go to two anonymous reviewers and Profs. George Perry and Min Zhu for constructive comments, which greatly improve the manuscript. This research has been supported by the National Key Research and Development Program of China (grant no. 2022YFF0802700), Chinese Academy of Sciences (grant no. 202200020), National Natural Science Foundation of China (grant no. 41720104002, 42072003, 42330209), Swedish Research Council (VR Project no. 2017-05183, 2018-03390, 2021-04295), and Zhongjian Yang Scholarship from the Department of Geology, Northwest University, Xi'an.

# Additional information

## Funding

| Funder | Grant reference number | Author |
| --- | --- | --- |
| National Key Research and Development Program of China | 2022YFF0802700 | Zhiliang Zhang |
| Chinese Academy of Sciences | 202200020 | Zhiliang Zhang |
| National Natural Science Foundation of China | 41720104002 | Zhifei Zhang |
| National Natural Science Foundation of China | 42072003 | Timothy P Topper |
| Swedish Research Council | 2017-05183 | Timothy P Topper |
| Swedish Research Council | 2018-03390 | Lars Holmer |
| Swedish Research Council | 2021-04295 | Timothy P Topper |
| Northwest University | Zhongjian Yang Scholarship | Lars Holmer |
| National Natural Science Foundation of China | 42330209 | Guoxiang Li |

The funders had no role in study design, data collection and interpretation, or the decision to submit the work for publication.

## Author contributions

Zhiliang Zhang, Conceptualization, Resources, Software, Funding acquisition, Validation, Investigation, Visualization, Methodology, Writing – original draft, Project administration, Writing – review and editing; Zhifei Zhang, Resources, Funding acquisition, Investigation, Writing – review and editing; Lars Holmer, Funding acquisition, Investigation, Writing – review and editing; Timothy P Topper, Guoxiang Li, Investigation, Writing – review and editing; Bing Pan, Writing – review and editing

## Author ORCIDs

Zhiliang Zhang ⓘ http://orcid.org/0000-0003-2296-5973
Zhifei Zhang ⓘ http://orcid.org/0000-0003-0325-5116
Timothy P Topper ⓘ http://orcid.org/0000-0001-6720-7418
Bing Pan ⓘ http://orcid.org/0000-0002-7457-2517

Reviewer #2 (Public Review): https://doi.org/10.7554/eLife.88855.4.sa1
Author response https://doi.org/10.7554/eLife.88855.4.sa2

---

# Additional files

## Supplementary files

• Supplementary file 1. Average dimensions and ratios of ventral and dorsal valves of *Latusobolus xiaoyangbaensis* gen. et sp. nov. from the Cambrian Series 2 Shuijingtuo Formation, South China.

• Supplementary file 2. Average dimensions and ratios of ventral and dorsal valves of *Eoobolus acutulus* sp. nov. from the Cambrian Series 2 Shuijingtuo Formation, South China.

• MDAR checklist

## Data availability

All data are available in the main text, the supplementary file, and source data materials.

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

# Appendix 1

## Systematic palaeontology

> Phylum Brachiopoda Duméril, 1806
> Subphylum Linguliformea Williams, Carlson, Brunton, Holmer and Popov, 1996
> Class Lingulata Gorjansky and Popov, 1985
> Order Lingulida Waagen, 1885
> Superfamily Linguloidea, Menke, 1828
> Family Eoobolidae, Holmer, Popov and Wrona, 1996
> Genus **Latusobolus** Zhang, Zhang and Holmer gen. nov.

## Type species

*Latusobolus xiaoyangbaensis* sp. nov., here designated.

## Etymology

From the Latin '*latus*' (wide) with the ending '*obolus*' (Greek coin), to indicate the transversely oval outline of both ventral and dorsal valves, morphologically similar to *Obolus*. The gender is masculine.

## Diagnosis

Shell transversely suboval or rounded triangular with gently straightened posterior margin, generally 108% wider than long, ventribiconvex. Apical angle is relatively large. Metamorphic shell outlined by a pronounced halo, and ornamented with evenly distributed pitted structures (pits), while post-metamorphic shell ornamented with finely concentric growth lines superposed with round pustules. Columnar shell structure. Ventral pseudointerarea orthocline with shallow and short pedicle groove. Propareas small, with weakly developed flexure lines, slightly raised up. Posterolateral muscle scars and *vascula lateralia* vestigial. Dorsal pseudointerarea orthocline. Propareas small, not raised. Median groove poorly defined, lacking flexure lines. Median tongue short. Umbonal muscle scars weakly impressed. Midian ridge and *vascula lateralia* vestigial.

## Remarks

The metamorphic pits and post-metamorphic pustules are very unique features for Eoobolidae (*Holmer et al., 1996*). Although having similar external ornamentation with contemporary *Eoobolus*, the new genus is established for the quite uncommon wide flat shape of both ventral and dorsal valves, while most *Eoobolus* possessing an elongate tongue shape. Compared to *Eoobolus*, the ventral pseudointerarea is smaller and gently raised, posterolateral muscles and pedicle nerve are weakly developed, and the pedicle groove is very short in *Latusobolus*. Furthermore, posterolateral muscles are vestigial, median tongue and median ridge are less developed in dorsal valve of the latter. As the columnar architecture is generally discovered in most well studied eoobolid brachiopods (*Holmer et al., 2008b*; *Streng et al., 2007*; *Ushatinskaya and Korovnikov, 2014*; *Zhang et al., 2020b*), it is very likely a new character for family Eoobolidae. However, the insufficient data of shell architecture in the holotype of *Eoobolus* indicates the comparison with the columnar shells in other eoobolids is not entirely unequivocal. Thus more thorough examinations on shell ultrastructures and ornamentation of early eoobolids and other linguliforms are needed in the future. This will be crucial for better understanding the evolution and phylogeny of brachiopods, and we hope that the comparison study undertaken here can further this aim.

> **Latusobolus xiaoyangbaensis** Zhang, Zhang, and Holmer sp. nov.
> *Figure 1*, *Appendix 1—figures 1–4*; *Supplementary file 1*.

## Etymology

After the occurrence at the Xiaoyangba section in southern Shaanxi, China.

## Holotype

ELI-XYB S5-1 BR09 (*Appendix 1—figure 1M–P*), ventral valve.

## Paratype

ELI-XYB S4-2 BO11 (*Appendix 1—figure 2M–P*), dorsal valve.

## Type locality

Cambrian Series 2 Shuijingtuo Formation (level S5-1) at the Xiaoyangba section (*Zhang et al., 2021a*) near Xiaoyang Village in Zhenba County, southern Shaanxi Province, China.

## Material

24 ventral and 17 dorsal valves, ranging from 624 µm to 2325 µm in length and from 669 µm to 2417 µm in width (*Appendix 1—figures 1 and 2* and *Supplementary file 1*) from Cambrian Series 2 Shuijingtuo Formation at the Xiaoyangba section, South China.

## Diagnosis

As for the genus.

## Description

Shell slightly ventribiconvex, transversely suboval to rounded triangular with gently acuminate posterior margin and rectimarginate anterior commissure, about 108% as wide as long (*Appendix 1—figures 1 and 2*). Metamorphic shell ornamented by regularly disposed hemispherical pits, uniform in size of about 0.5 µm, ranging from 0.3 µm to 0.7 µm in diameter (*Appendix 1—figure 3H* and *Supplementary file 1*). The pronounced halo marks the boundary between metamorphic shell and post-metamorphic shell (*Appendix 1—figure 3A–E*). A narrow belt of about 50 µm width, outside the metamorphic shell lacks pustules and may belong to the neanic shell (*Appendix 1—figure 3A–D*). Post-metamorphic shell fully covered with fine concentric growth lines superposed by finely pustular ornamentation, (*Appendix 1—figures 1I and 2M*). Round pustules have mean diameter of 6.5 µm, ranging from 2.3 µm to 12.6 µm (*Appendix 1—figure 3F and G*), packed in concentric rows and rarely in radial pattern (*Appendix 1—figure 3F*).

Ventral valve obtuse with apical angle of 129° on average, rounded triangular, and gently convex in sagittal profile, about length 95% width and depth 18% length, with maximum width slightly anterior to mid-length and maximum height slightly posterior to mid-length (*Supplementary file 1*). Pseudointerarea, small, orthocline, occupying 11% valve length and 44% valve width, with a shallow, short, subtriangular pedicle groove of about 112 µm in length and 120 µm in width, with about 64% length of the pseudointerarea (*Appendix 1—figure 4A and C–E*). Lateral sides of the pedicle groove divergent anteriorly at about 34°. Propareas narrow, raised above the valve floor, divided into two parts by shallow flexure lines, while the internal part of proparea is slightly larger than the external part (*Appendix 1—figure 1J*; *Appendix 1—figure 4A*). Ventral valve interior with a vestigial visceral area, bisected by the divergent pedicle nerve impression terminated anteriorly at about 36% of valve length (*Appendix 1—figure 4E*). Paired posterolateral muscle scars weakly developed underneath the raised propareas (*Appendix 1—figure 4A and E*). *Vascula lateralia* weakly developed only on large valves, submarginal, divergent proximally (*Appendix 1—figure 1O* and *Appendix 1—figure 4A*). Metamorphic shell 223 µm long and 275 µm wide on average, bounded by a pronounced halo (*Appendix 1—figure 3A–C*). Possible protegulum noted as a distinct mound about 50 µm across located posteromedially and bearing faint folds anterolaterally (*Appendix 1—figure 3B*).

Dorsal valve obtuse with apical angle of 136° on average, rounded triangular, and slightly convex in sagittal profile, about 93% shorter than wide and depth 20% length, with maximum width slightly anterior to mid-length and maximum height slightly posterior to mid-length (*Supplementary file 1*). Pseudointerarea, small, orthocline, occupying 9% valve length and 41% valve width, with a shallow, short, subtriangular median groove of about 104 µm in length and 292 µm in width, with about 83% length of the pseudointerarea (*Appendix 1—figure 4F and G*). Median groove divergent anteriorly with an average angle of 112°. Propareas narrow, not raised above the valve floor. Dorsal valve interior with a vestigial visceral area. Median ridge narrow, weakly developed on large valves (*Appendix 1—figure 4G and H*). Paired posterolateral muscle scars weakly developed anterolaterally to the median groove (*Appendix 1—figure 2N*; *Appendix 1—figure 4G*). Metamorphic shell 198 µm long and 260 µm wide on average, bounded by a pronounced halo (*Appendix 1—figure 3D and E*). Possible protegulum noted as a distinct mound about 50 µm across located posteromedially and bearing faint folds anterolaterally (*Appendix 1—figure 3E*).

Shell structure stratiform, consisting of laminated primary layer and columnar secondary layer (*Figure 1*). The laminated primary layer is composed of compact apatitic lamellae, about 3 μm thick (*Figure 1B*), while the secondary layer is composed of stacked sandwich columnar units (totalled 1–3 units), including numerous columns disposed orthogonally between a pair of stratiform lamellae (*Figure 1B and C*; *Appendix 1—figure 4I and J*). The hollow space in the columns and between lamellae of stacked columnar units probably indicates the rich composition of organic material (*Figure 1C and D*; *Appendix 1—figure 4I*). Columns are quite small about 2.4 μm in diameter, ranging from 1.6 μm to 3.4 μm, and about 6 μm in height, ranging from 2.9 μm to 11.9 μm. The central canal in the column small, ranging from 0.4 μm to 0.9 μm in diameter. The central space between the stratiform lamellae, thin, around 0.7 μm, while the stratiform lamellae of columnar units about 1.4 μm in thickness (*Figure 6—source data 1*).

> Genus *Eoobolus* Matthew, 1902
> Type species. *Obolus* (*Eoobolus*) *triparilis* Matthew, 1902 (selected by *Rowell, 1965*)
> Diagnosis. See Holmer et al. (p. 41) (*Holmer et al., 1996*).
> *Eoobolus acutulus* Zhang, Zhang, and Holmer sp. nov.
> *Figure 2* and *Appendix 1—figures 5–7*, *Supplementary file 2*.

## Etymology

From the Latin '*acutulus*' (somewhat pointed), to indicate the slightly acuminate ventral valve with an acute apical angle. The gender is masculine.

## Holotype

ELI-AJH S05 BT11 (*Appendix 1—figure 5E–H*), ventral valve.

## Paratype

ELI-AJH S05 1-5-07 (*Appendix 1—figure 5M*), dorsal valve.

## Type locality

Cambrian Series 2 Shuijingtuo Formation (level S05) at the Aijiahe section (*Zhang et al., 2016b*) near Aijiahe Village in Zigui County, north-western Hubei Province, China.

## Material

20 ventral and three dorsal valves, ranging from 715 μm to 1877 μm in length and from 562 μm to 1466 μm in width (*Appendix 1—figure 5* and *Supplementary file 2*) from Cambrian Series 2 Shuijingtuo Formation at the Aijiahe section, South China.

## Diagnosis

Shell elongate oval with acuminate posterior margin, generally 128% longer than wide, biconvex. Metamorphic shell outlined by a pronounced halo, and ornamented with evenly distributed pits, while post-metamorphic shell ornamented with finely circular growth lines and large random pustules. Columnar shell structure. Ventral pseudointerarea orthocline with steep and narrow pedicle groove. Propareas small, with flexure lines. Posterolateral, umbonal and anterior muscle scars, and pedicle nerve weakly impressed. *Vascula lateralia* indefinite. Dorsal pseudointerarea orthocline. Propareas not raided. Median groove wide, poorly defined laterally, lacking flexure lines, short. Dorsal visceral area weakly impressed.

## Description

Shell slightly ventribiconvex, elongate oval with acuminate posterior margin and rectimarginate anterior commissure, about 128% as long as wide (*Appendix 1—figure 5*). Metamorphic shell ornamented by regularly disposed hemispherical pits, uniform in size of about 0.6 μm, ranging from 0.2 μm to 1.1 μm in diameter (*Appendix 1—figure 6F–H* and *Supplementary file 2*). The pronounced halo marks the boundary between metamorphic shell and post-metamorphic shell (*Appendix 1—figure 6A and B*). Neanic shell indefinite. Post-metamorphic shell fully covered with finely and densely concentric growth lines superposed by elongate pustulose ornamentation, (*Appendix 1—figure 5C*, *Appendix 1—figure 6J and K*). Elongate pustules have mean length of

11.8 μm, ranging from 8 μm to 21 μm, and mean width of 4.7 μm (*Supplementary file 2*), packed in concentric rows.

Ventral valve, acuminate with an acute apical angle of 83° on average ranging from 73° to 89°, elongate oval, and gently convex in sagittal profile, about length 130% width and depth 21% length, with maximum width slightly anterior to mid-length and maximum height slightly posterior to mid-length (*Supplementary file 2*). Pseudointerarea, orthocline, occupying 25% valve length and 69% valve width, with a deep, narrow, subtriangular pedicle groove of about 341 μm in length and 251 μm in width, with about 43% length of the pseudointerarea (*Appendix 1—figure 7A–C and G*). Lateral sides of the pedicle groove slightly divergent anteriorly at about 17°. Propareas narrow, raised above the valve floor, divided into two parts by robust flexure lines, while the internal part of proparea is wider than the external part (*Appendix 1—figure 6A and F*). Ventral valve interior with a vestigial visceral area, bisected by the divergent pedicle nerve impression terminated anteriorly at about 46% of valve length (*Appendix 1—figure 7C*). Paired posterolateral muscle scars weakly developed underneath the raised propareas (*Appendix 1—figure 7B*). Anterior and umbonal muscle scars weakly developed only on large valves (*Appendix 1—figure 7A and C*). *Vascula lateralia* indefinite. Metamorphic shell 229 μm long and 255 μm wide on average, bounded by a pronounced halo (*Appendix 1—figure 6A–D*). Possible protegulum noted as a distinct mound about 50 μm across, located posteromedially and bearing faint folds anterolaterally (*Appendix 1—figure 6C*).

Dorsal valve with apical angle of 90° on average, elongate oval, about 128% longer than wide, with maximum width slightly anterior to mid-length (*Supplementary file 2*). Pseudointerarea, wide, orthocline, occupying 19% valve length and 67% valve width, with a wide, shallow, subtriangular median groove of about 165 μm in length, with about 75% length of the pseudointerarea (*Appendix 1—figure 7H*). Median groove outline poorly diagnosed on pseudointerarea. Propareas very narrow. Dorsal valve interior with a vestigial visceral area. Median ridge vestigial (*Appendix 1—figure 5M*). Muscle systems poorly diagnosed.

Shell structure stratiform, consisting of laminated primary layer and columnar secondary layer (*Figure 2*). The laminated primary layer is composed of compact apatite, about 1.5 μm thick (*Figure 2G*), while the secondary layer is composed of multi-stacked sandwich columnar units (maximum 13 units), including numerous columns disposed orthogonally between a pair of stratiform lamellae (*Figure 2B, C, J and K*; *Appendix 1—figure 7J*). The hollow space in the columns and between stratiform lamellae of columnar units probably indicates the rich composition of organic material (*Figure 2F–H*). Columns are quite small about 2.4 μm in diameter, ranging from 1.2 μm to 3.2 μm, and about 4 μm in height, ranging from 1.7 μm to 6.5 μm. The central canal in the column small, ranging from 0.4 μm to 1 μm in diameter. The space between stratiform lamellae of stacked columnar units, thin, around 0.6 μm, while the stratiform lamellae about 1.2 μm in thickness (*Figure 6—source data 1*).

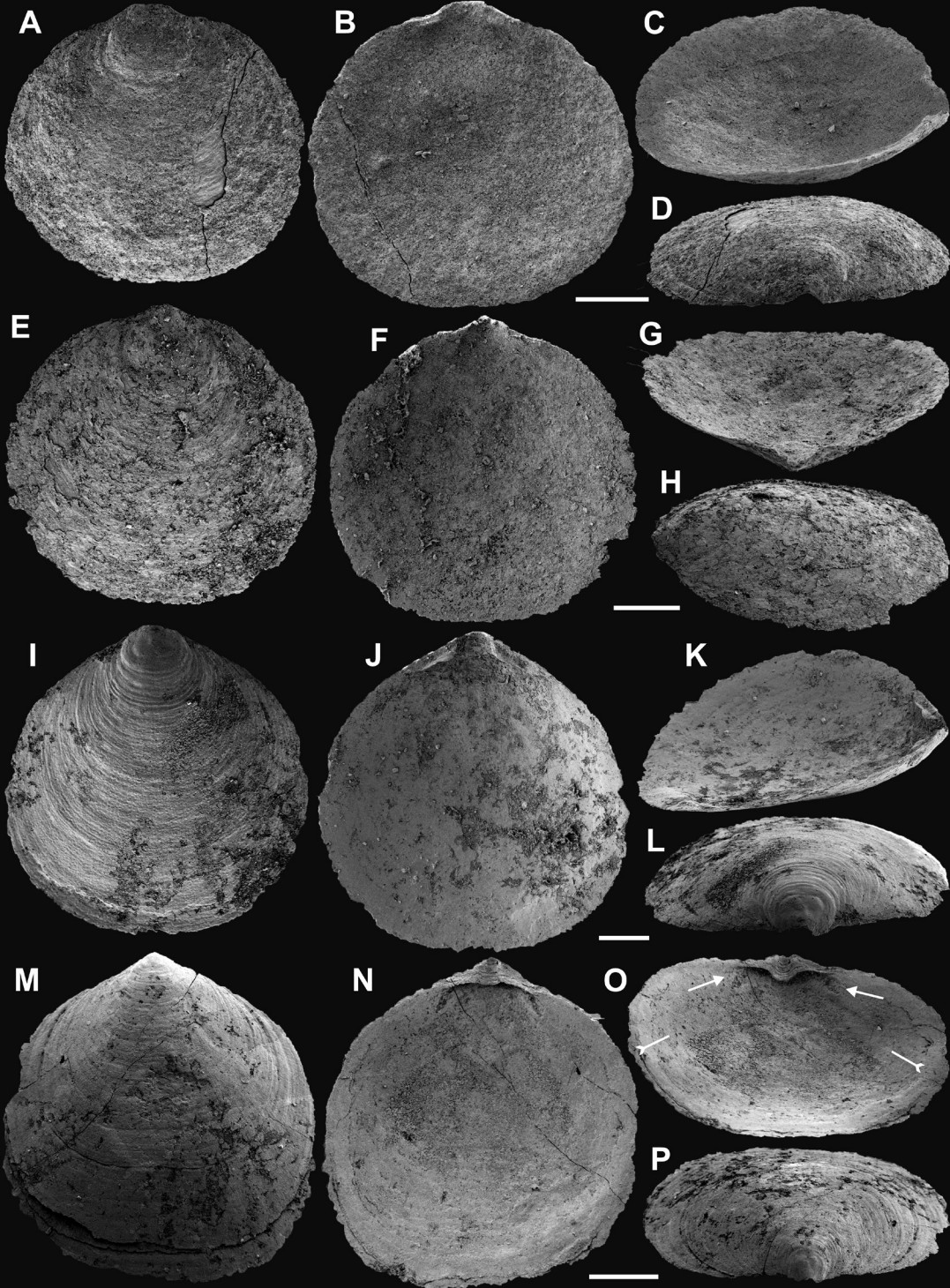

**Appendix 1—figure 1.** Ventral valves of *Latusobolus xiaoyangbaensis* gen. et sp. nov. from the Cambrian Series 2 Shuijingtuo Formation in southern Shaanxi, South China. (**A–D**) Juvenile with weakly developed pseudointerarea, ELI-XYB S5-1 BR01. (**E–H**) Juvenile with weakly developed pseudointerarea, ELI-XYB S5-1 BR02. (**I–J**) Large valve with slightly elevated pseudointerarea, ELI-XYB S5-1 BR04. (**M–P**) Mature valve with developed posterolateral muscle scars (arrows) and rudiment of *vascula lateralia* (tailed arrows), ELI-XYB S5-1 BR09. Scale bars: (**A–L**), 200 µm; (**M–P**), 500 µm.

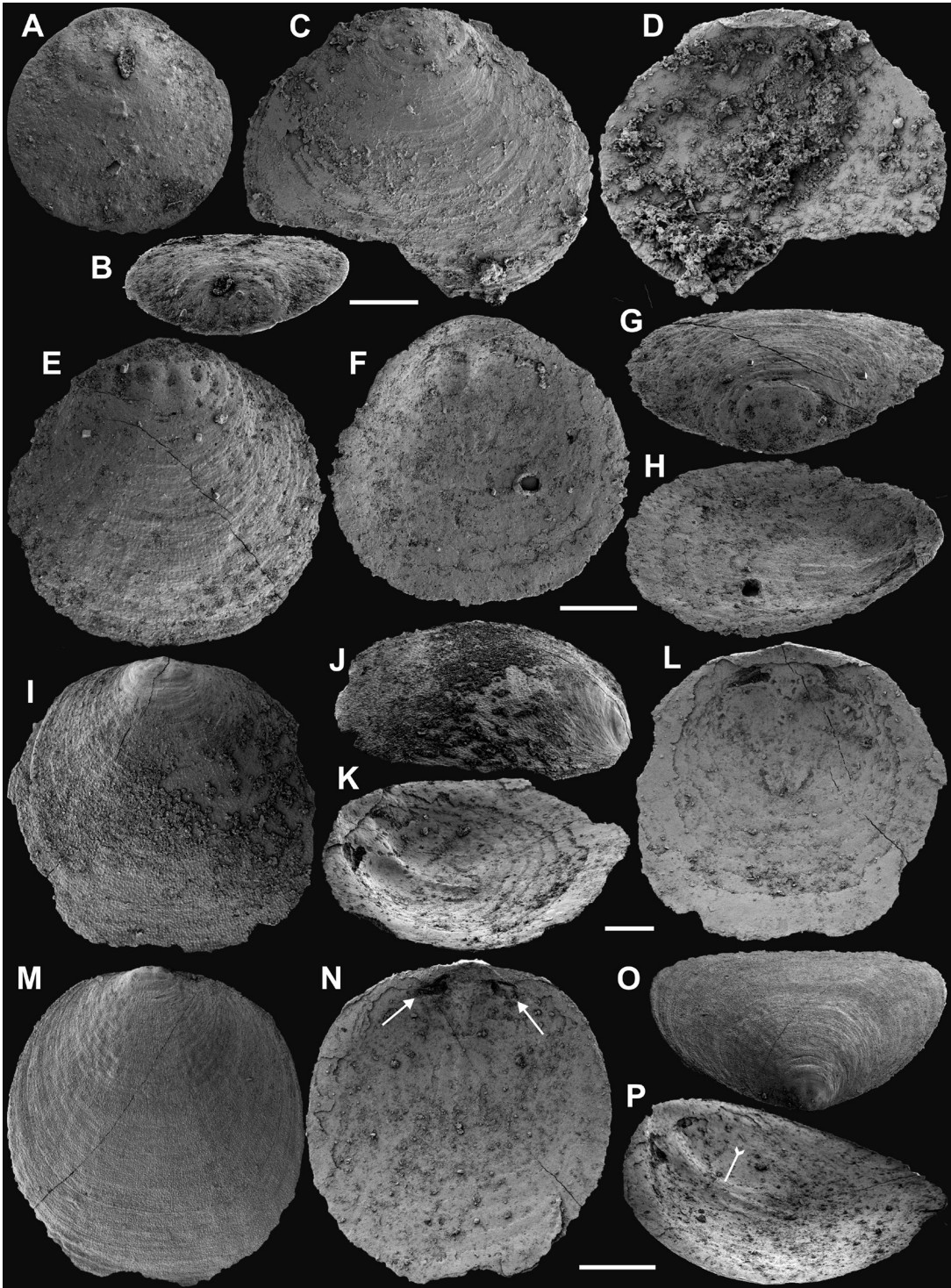

**Appendix 1—figure 2.** Dorsal valves of *Latusobolus xiaoyangbaensis* gen. et sp. nov. from the Cambrian Series 2 Shuijingtuo Formation in southern Shaanxi, South China. (**A–B**) Small juvenile with weakly developed pustular ornamentation, ELI-XYB S5-1 BS15. (**C–D**) Juvenile, ELI-XYB S4-2 BO08. (**E–H**) Juvenile with rudiment of median ridge, ELI-XYB S4-2 BS09. (**I–L**) Large valve, ELI-XYB S4-2 BO13. (**M–P**) Mature valve with weakly developed paired posterolateral muscle scars (arrows), median ridge and pair of submedian ridges bisecting dorsal visceral field (tailed arrow), ELI-XYB S4-2 BO11. Scale bars: (**A–D**), 100 μm; (**E–L**), 200 μm; (**M–P**), 500 μm.

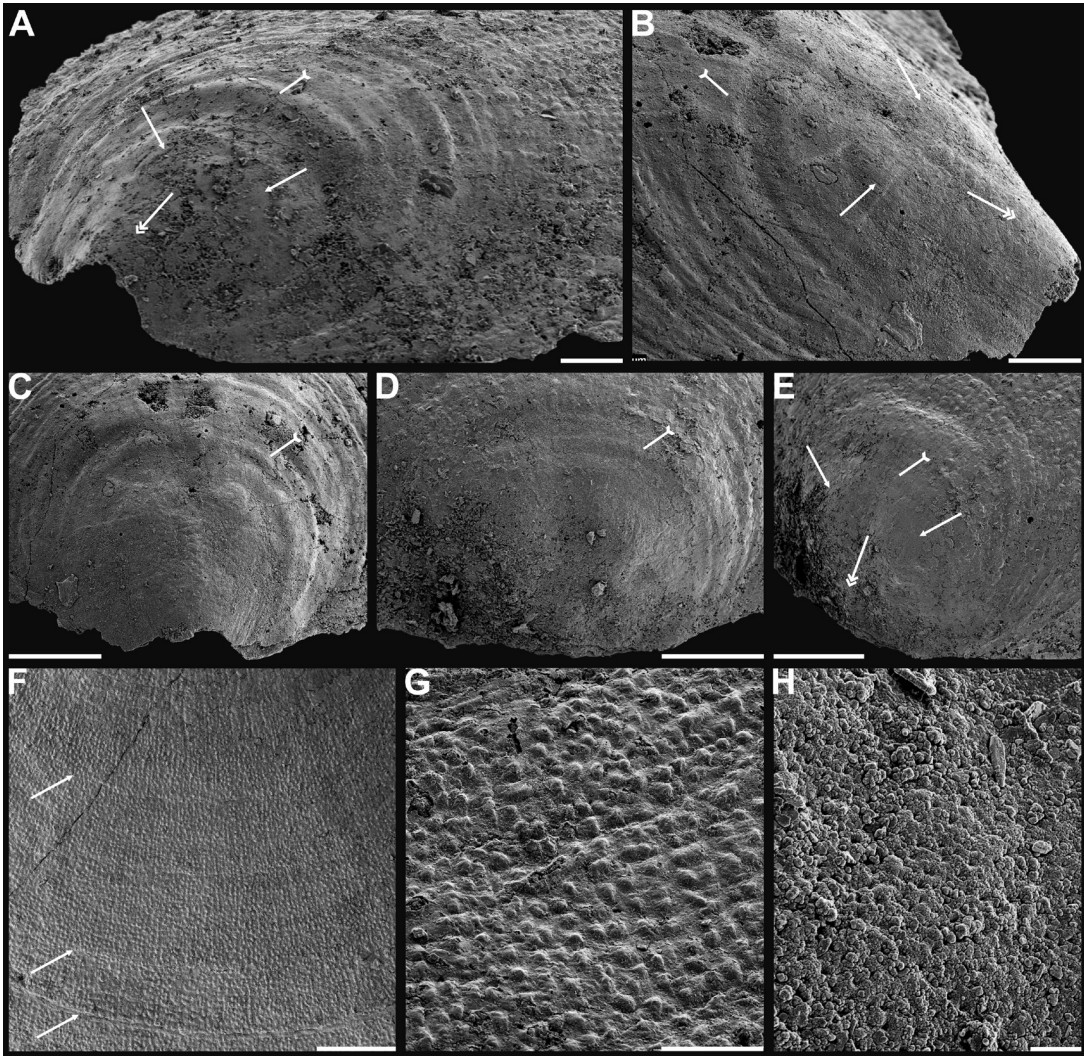

**Appendix 1—figure 3.** Shell characters and ornamentation of *Latusobolus xiaoyangbaensis* gen. et sp. nov. from the Cambrian Series 2 Shuijingtuo Formation in southern Shaanxi, South China. (**A**) Enlarged ventral metamorphic shell, noting the developed halo by tailed arrow, protegulum by double-headed arrow and brephic lobes by arrows, ELI-XYB S4-3 AU10. (**B–C**) Ventral valve, ELI-XYB S4-2 BO09. (**B**) Metamorphic shell of mature valve, noting the halo by tailed arrow, protegulum by double-headed arrow and brephic lobes by arrows. (**C**) Posterior view, noting the halo by tailed arrow. (**D–G**) Dorsal valve, ELI-XYB S4-2 BO11. (**D**) Metamorphic shell, noting the halo by tailed arrow. (**E**) Lateral dorsal view, noting the halo by tailed arrow, protegulum by double-headed arrow and brephic lobes by arrows. (**F**) Post-metamorphic pustules, note sparsely packed concentric growth lines by arrows. (**G**) Enlarged pustules. (**H**) Enlargement of metamorphic pitted ornament, ELI-XYB S4-2 BO08. Scale bars: (**A**), (**B**), (**G**), 50 µm; (**C–E**), 100 µm; (**F**), 200 µm; (**H**), 5 µm.

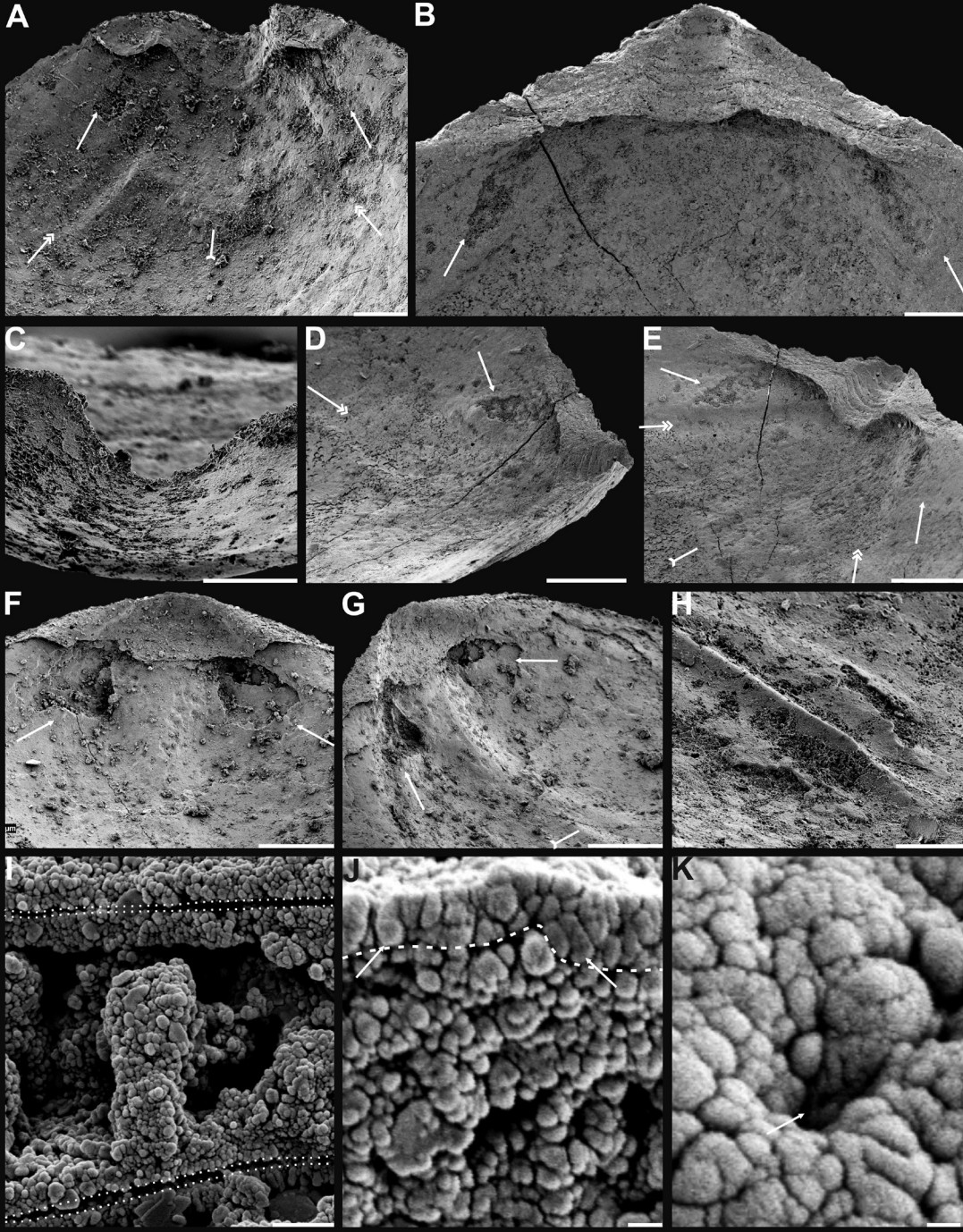

**Appendix 1—figure 4.** Internal characters and shell ultrastructures of *Latusobolus xiaoyangbaensis* gen. et sp. nov. from the Cambrian Series 2 Shuijingtuo Formation in southern Shaanxi, South China. (**A–B**) Ventral valve, ELI-XYB S4-3 AU10. (**A**) Interior showing posterolateral muscle scars below propareas by arrows, pedicle nerve by tailed arrow, *vascula lateralia* by double-headed arrows. (**B**) Posterior view of pedicle groove. (**C–E**) Ventral valve, ELI-XYB S5-1 BR09. (**C**) Interior view showing triangular pseudointerarea and paired posterolateral muscle scars by arrows. (**D–E**) Lateral view, note elevated pseudointerarea with posterolateral muscles underneath by arrows, pedicle nerve by tailed arrow, *vascula lateralia* by double-headed arrows. (**F–G**) Dorsal valve, noting paired umbonal muscle scars by arrows, median ridge by tailed arrow, ELI-XYB S4-2 BO11. (**H**) Enlargement of terminal of median ridge, ELI-XYB S5-1 BS07. (**I**) Ventral, enlarged columnar architecture, outlining organic membranes between the stratiform lamellae of stacked sandwich columnar units, ELI-XYB S5-1 BS01. (**J**) Nanoscale apatite spherules of granule aggregations (arrows) of ventral shell, note primary-secondary layer boundary by dashed line, ELI-XYB S4-2 BO12. (**K**) Enlarged nanoscale spherules of pitted ornament (arrow) on primary layer, ELI-XYB S4-2 BO08. Scale bars: (**A**), (**B**), (**H**), 50 μm; (**C**), 100 μm; (**D–G**), 200 μm; (**I**), 2 μm; (**J**), 500 nm; (**K**), 200 nm.

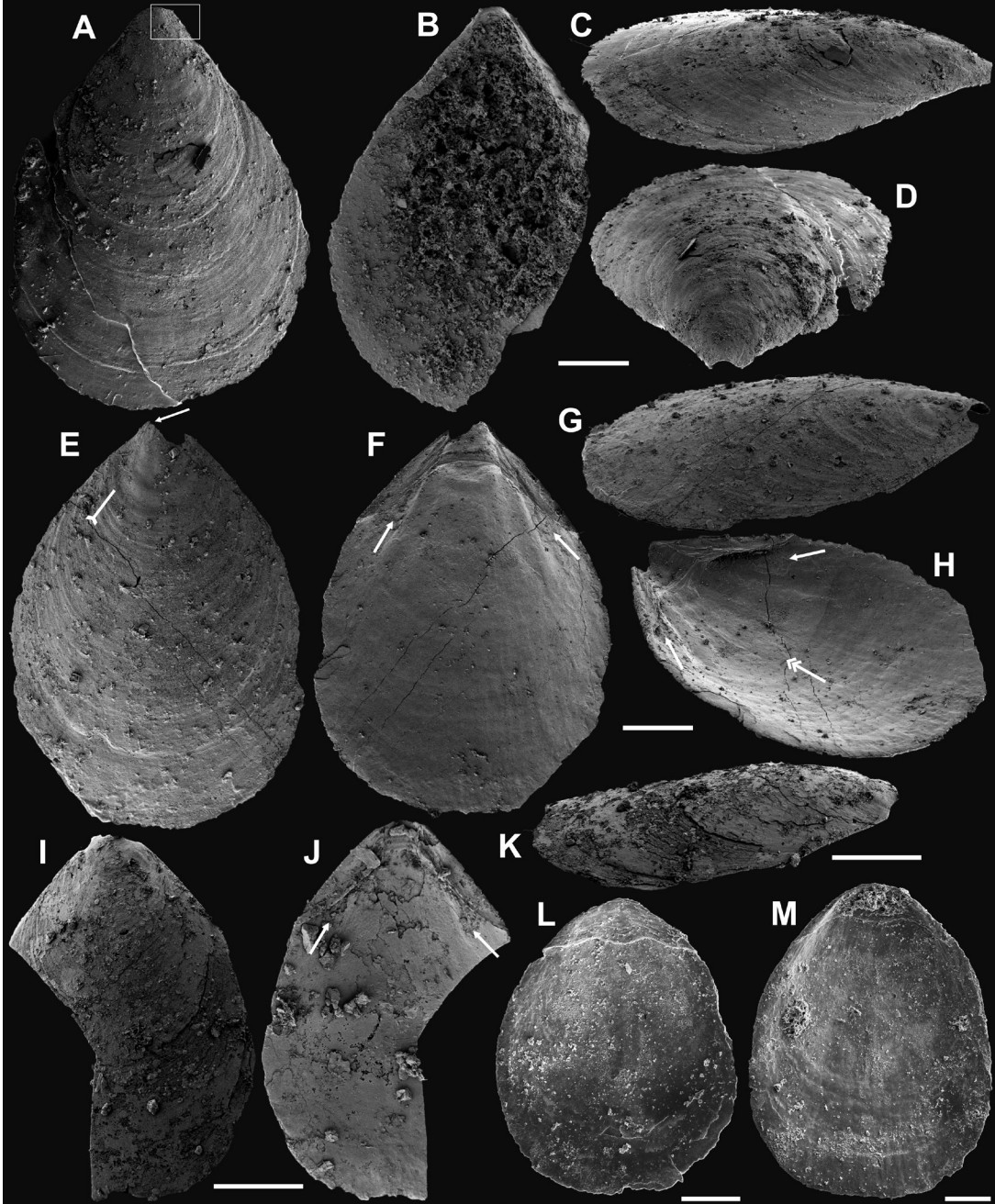

**Appendix 1—figure 5.** Ventral and dorsal valves of *Eoobolus acutulus* sp. nov. from the Cambrian Series 2 Shuijingtuo Formation in Three Gorges areas, South China. (**A–D**) Juvenile with weakly developed pseudointerarea, box indicates area in ***Appendix 1—figure 6D***, ELI-WJP 7 CE05. (**E–H**) Juvenile, note slightly developed pseudointerarea, the halo by tailed arrow, paired posterolateral muscle scars by arrows, pedicle nerve by double-headed arrow, ELI-AJH S05 BT11. (**I–K**) Mature valve, developed paired posterolateral muscle scars (arrows) underneath elevated pseudointerarea, ELI-AJH S05 BT14. (**L–M**), Dorsal valve with rudiment of median ridge, ELI-AJH 1-5-01, ELI-AJH 1-5-07. Scale bars: (**A–H**), (**L**), (**M**), 200 µm; (**I–K**), 400 µm.

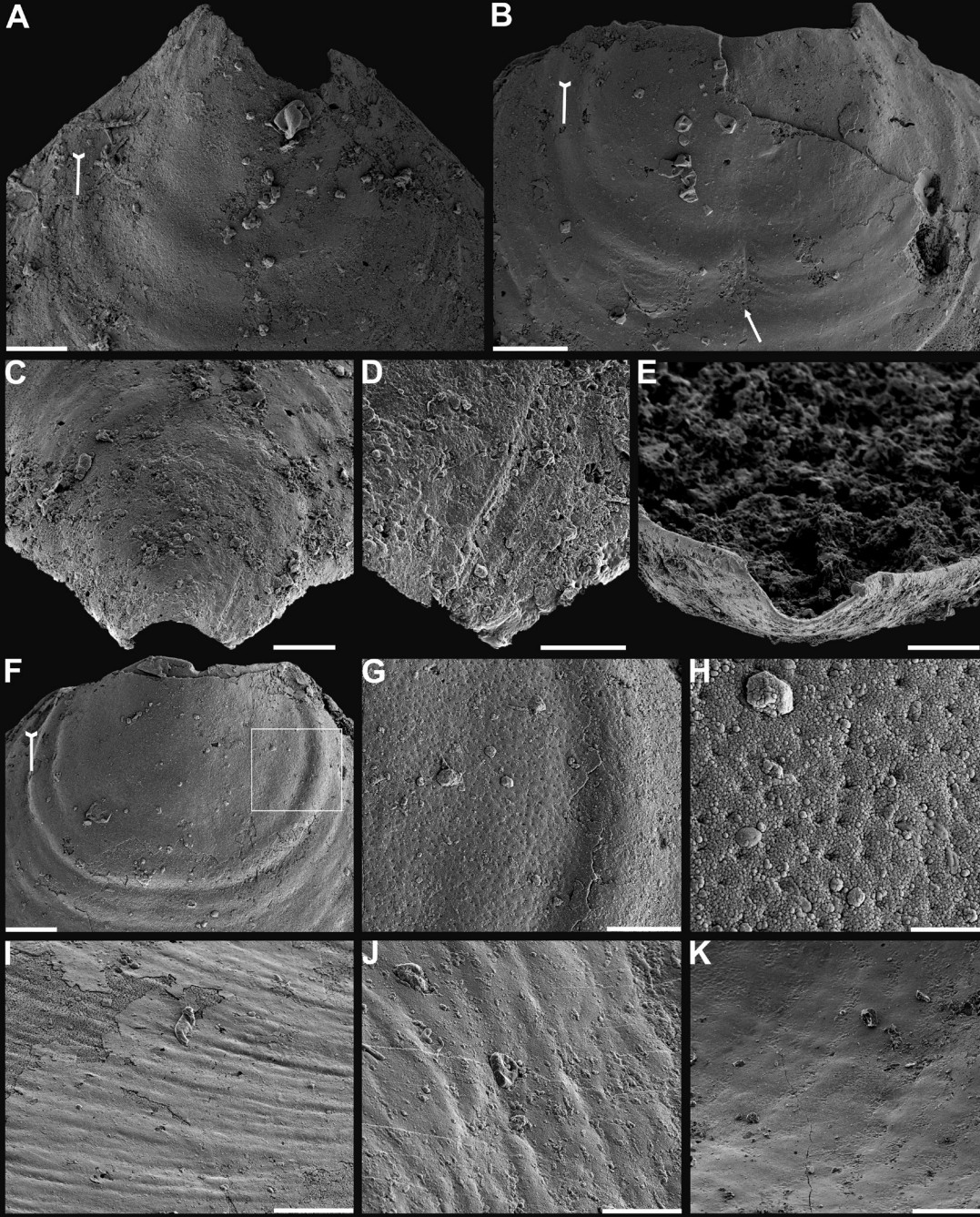

**Appendix 1—figure 6.** Shell characters and ornamentation of *Eoobolus acutulus* sp. nov. from the Cambrian Series 2 Shuijingtuo Formation in Three Gorges areas, South China. (**A**) Enlarged ventral metamorphic shell, noting the halo by tailed arrow, ELI-AJH S05 BT11. (**B**) Enlarged ventral metamorphic shell, noting the halo by tailed arrow and drape structures outside the halo by arrow, ELI-AJH S05 BT05. (**C–E**) Ventral valve, ELI-WJP 7 CE05. (**C**) Posterior ventral view of metamorphic shell. (**D**) Fine ridges on the margin of protegulum. (**E**) Posterior view of pedicle groove. (**F–H**) Ventral valve, ELI-AJH ST 8-2-3 BT04. (**F**) Metamorphic shell, noting the halo by tailed arrow, box indicates area in **G**. (**G**) Metamorphic pits. (**H**) Enlarged pits. (**I**) Dense concentric growth lines on ventral external surface, ELI-AJH 8-2-3 BT04. (**J**) Elongate pustular ornamentation on external surface, ELI-AJH 8-2-3 BT03. (**K**) Elongate pustule ornamentation on interior, ELI-AJH 8-2-3 BT04. Scale bars: (**A–C**), (**E**), (**F**), (**I**), (**K**), 50 μm; (**D**), (**G**), 20 μm; (**H**), 5 μm; (**J**), 10 μm.

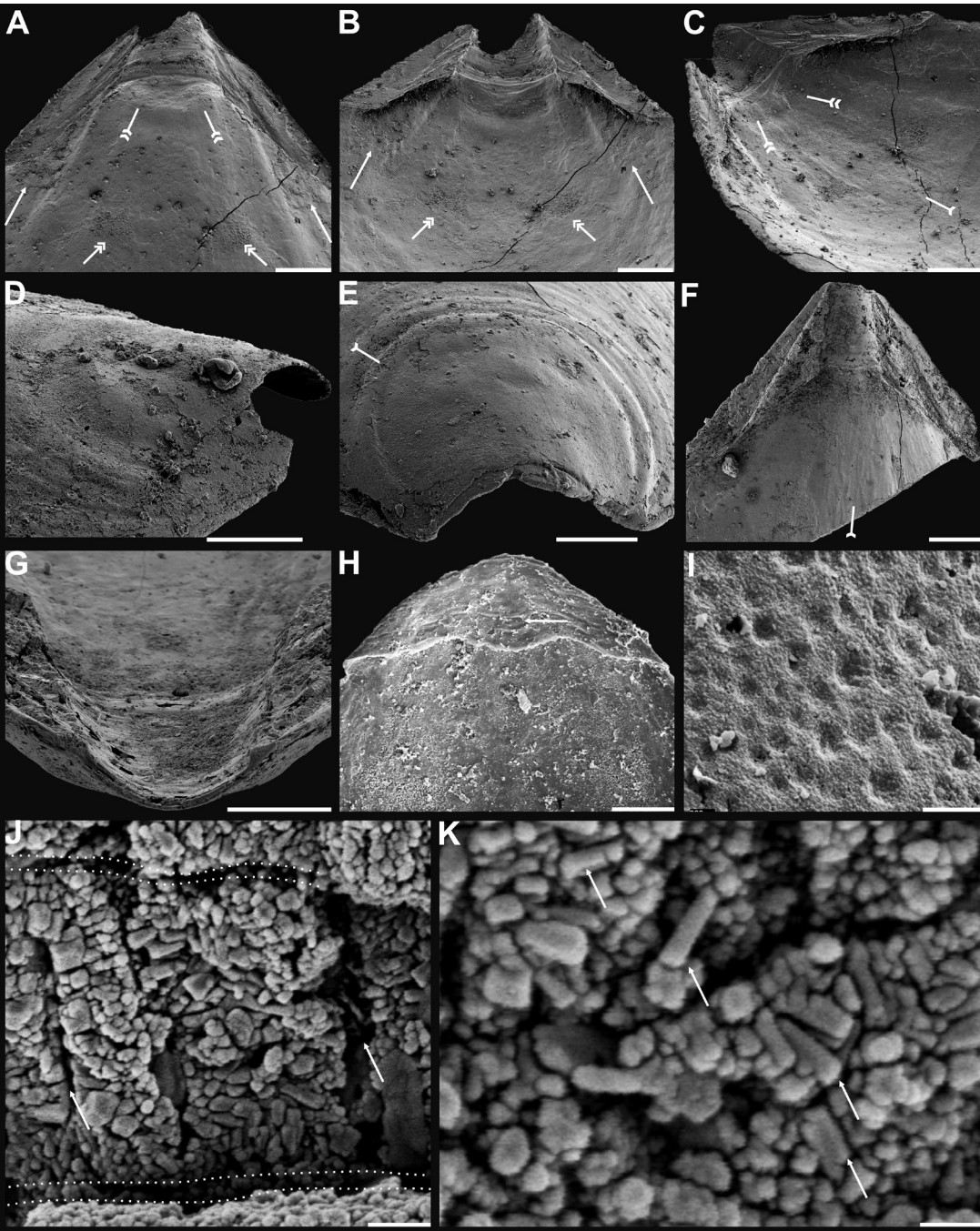

**Appendix 1—figure 7.** Shell characters and ultrastructures of *Eoobolus acutulus* sp. nov. from the Cambrian Series 2 Shuijingtuo Formation in Three Gorges areas, South China. (**A–D**) Ventral valve, ELI-AJH SJT S05 BT11. (**A–C**) Interior noting elevated pseudointerarea with posterolateral muscles underneath by arrows, pedicle nerve by tailed arrow, anterior muscle scars by double-headed arrows, umbonal muscle scars by double-tailed arrows. (**D**) Lateral view of metamorphic shell. (**E**) Posterior view of ventral metamorphic shell, note the halo by tailed arrow, ELI-AJH SJT S05 BT04. (**F–G**) Ventral valve, ELI-AJH SJT 8-2-3 BT02. (**F**) Pseudointerarea of a large valve, noting pedicle nerve by tailed arrow. (**G**) Posterior view of pedicle groove. (**H**) Dorsal pseudointerarea, ELI-AJH 1-5-01. (**I**) Enlarged pitted ornament on primary layer, ELI-AJH 8-2-2 Lin005. (**J–K**) Ventral valve, ELI-AJH S05 BT12. (**J**) Apatite spherules of granule aggregations, note organic canals of columns by arrows, outlining organic membranes between the stratiform lamellae of stacked sandwich column units. (**K**) Enlarged nanoscale spherules of granule aggregations, note elongate rods by arrows. Scale bars: (**A–C**), (**G**), (**H**), 100 µm; (**D**), 50 µm; (**E**), (**F**), 200 µm; (**I**), 2 µm; (**J**), 1 µm; (**K**), 400 nm.

