## [Editor Report · eLife assessment]

This **valuable** study examines the evolution of the pillars in the shell architecture of organo-phosphatic brachiopods. The phylogenetic implications of this shell structure in relation to other early Cambrian brachiopod families are interpreted based on **solid** evidence. As such, this paper with interesting ideas regarding the evolution of brachiopod shell structure contributes to our understanding of the ecology and evolution of brachiopods as a whole.

---

## [Referee Report · Reviewer #2 (Public Review)]

Summary:

Two early Cambrian taxa of linguliform brachiopods are assigned to the family Eoobolidae. The taxa exhibit a columnar shell structure and the phylogenetic implications of this shell structure in relation to other early Cambrian families is outlined.

Strengths:

Interesting idea regarding the evolution of shell structure.

Weaknesses:

The early record of shell structures of linguliform brachiopods is incomplete and partly contradictory. The authors maintain silence regarding contradictory information throughout the article to an extend that information is cited wrongly. The article is written under the assumption that all eoobolids have a columnar shell structure. Thus, the previously claimed columnar structure of Eoobolus incipiens which has been re-illustrated in the paper is not convincing and could be interpreted in other ways.

The article still needs a proper results section. The Discussion is mainly a review of published data. Other potential results are hidden in this "discussion". Large sections of the paper appear irrelevant and can be deleted.

A critical revision of the family Eoobolidae and Lingulellotretidae including a revision of the type species of Eoobolus and Lingulellotreta is needed first.

The potential evolutionary patterns that are presented towards the end (summarized in Fig 6) are interesting but rather unconvincing. The stated numbers of shell laminae, whose origin has now been clarified in a still rather short Methods section, represent a mix of data and are not comparable. Achieved numbers of laminae based on literature data include laminae from the secondary and tertiary shell layer, a distinction between the two would be crucial for the proposed claims.

The obtained evolutionary patterns as presented in Fig. 6 must, after the second revision and clarification of the methods used, be regarded as misleading and reflects a limited understanding of shell growths in linguliform brachiopods (despite the extensive review of the literature).

---

## [Author Response]

The following is the authors’ response to the previous reviews.

We greatly appreciate your positive assessment and the suggestions by Reviewer #2 on the previous version of our manuscript, all of which are very helpful and have greatly improved our manuscript. We have added a description of Biomineralized columnar architecture in the Results section, added a discussion of the Family Eoobolidae, provided more details in the Material and Methods section, and revised other parts of the manuscript based on her/his comments. We are grateful that these comments have enhanced the overall quality of our manuscript. In this letter, we take the opportunity to note and discuss the various changes as below.

**Reviewer #2:**
(1) Two early Cambrian taxa of linguliform brachiopods are assigned to the family Eoobolidae. The taxa exhibit a columnar shell structure and the phylogenetic implications of this shell structure in relation to other early Cambrian families is discussed. It is the interesting idea regarding the evolution of shell structure.

We thank Reviewer 2 very much for her/his very constructive suggestions. All the comments have been thoroughly considered, and introduced into the revised version of the manuscript.

(2) The early record of shell structures of linguliform brachiopods is incomplete and partly contradictory. The authors maintain silence regarding contradictory information throughout the article to an extent that information is cited wrongly.

We agree with Reviewer #2 that the early record of shell structure of linguliform brachiopods is incomplete and potentially in some instances contradictory. This situation is well demonstrated in the Introduction and Systematic Palaeontology sections of this paper. This is also the reason why we think the detailed investigation of early linguliform shell architectures is so important, and we hope this work will be useful for further comparative studies on brachiopod biomineralization. We also understand that more detailed studies of the complexity and diversity of linguliform brachiopod architectures (especially their early fossil representatives) require further investigation.

(3) The article is written under the assumption that all eoobolids have a columnar shell structure. Thus, the previously claimed columnar structure of Eoobolus incipiens which has been re-illustrated in the paper is not convincing and could be interpreted in other ways.

Yes, the type specimen of Eoobolus is poorly known and we do not know its shell structure, but the ornamentation, pseudointerarea etc. are well preserved and promote a character diagnosis. In this paper, we focus on the detailed study of Cambrian eoobolids with exquisitely well-preserved columns from the Cambrian Series 2 based on the collection of more than 30 thousand early Cambrian brachiopod specimens in China and Australia. With the wide preservation of columnar shells in early eoobolid specimens, it is likely that Eoobolus has columnar shell architecture, although there is no documentation of the shell structure from every single Eoobolus specimen.

The secondary columns of Eoobolus incipiens is well demonstrated in Fig. 4a. The size of these columns can be well compared with the columns from other Eoobolus species and acrotretide brachiopods, which are quite different from the criss-cross baculae. As we noted in the manuscript, the columnar structure Eoobolus incipiens is very simple (short columns and less number of columnar units) and can be readily secondarily phosphatised. This is also the reason why it is hard to find the columnar shell architecture in early eoobolids.

(4) The article needs a proper results section. The Discussion is mainly a review of published data. Other potential results are hidden in this "discussion".I would recommend to reorganize the paper and make it a solid presentation of the new taxa and other new results, i.e., have a solid Results section. The Discussion should discuss relevant points that relate to the new results rather than reviewing shell structure in general but skipping relevant parts such as the tertiary shell layer.

We have reorganised the manuscript based on these comments. A general description of the biomineralized columnar architecture is added in the Results section. As the Supplementary section (main results) includes 7 figures and 3 tables, it will increase the size of the current paper if they are moved to the main text. We would prefer to keep the main results in the Supplementary based on the style and format of eLife.

As the current information on the shell structures of early linguliform brachiopods is unclear, we need to review most of the previous studies on brachiopod shells in the first part of Discussion section. It will help the readers to follow our results and conclusion. So, we think some of the review content is necessary and helps build the Discussion section. The tertiary shell layer, which is not developed in our studied material, is not discussed in the current research.

(5) In addition, a more elaborate Methods section is needed in which it is explained how the data for shell thicknesses and numbers of laminae was obtained.The potential evolutionary patterns that are discussed towards the end (summarized in Fig 6) are interesting but rather unconvincing as the way the data has been obtained has never been clarified. Shell thicknesses and numbers of laminae that built up the shell of several taxa are compared, but at no point it is stated where these measurements were taken. Shell thicknesses vary within a shell and also the presence of the never mentioned tertiary layer is modifying shell thicknesses. Hence, the presented data appears random and is not comparable. The obtained evolutionary patterns must be considered as dubious.A proper Methods section would be needed that explains how the data presented in Fig. 6 has been obtained. Plus it needs to be convincingly explained that the obtained data is in fact comparable and represents, e.g., equivalent areas of the shell in all involved taxa.

All the information is added in the Material and Methods section. We are aware of the marginal accretionary secretion of brachiopod shells. It is well known that the shell at the posterior is thicker (usually the thickest) than that at the anterior, we did not note this in the previous manuscript. We have measured all the shell data (shell thickness and number of columnar unit) from the posterior part of the adult shell for all the studied taxa. And the measurements of diameter and height of orthogonal columns are performed on available adult specimens from this study and previously published literature. Consequently, the obtained data are comparable and represent equivalent areas of the shell on all involved taxa.

In term of the tertiary shell layer, we do not find any evidence of this tertiary shell layer from our studied material. The tertiary shell layer is well developed in some recent and Palaeozoic lingulides (Holmer, 1989), but it is not recognised in the early eoobolides and acrotretides.

(6) A critical revision of the family Eoobolidae and Lingulellotretidae including a revision of the type species of Eoobolus and Lingulellotreta is needed.

Concerning the families Eoobolidae and Lingulellotretidae, we are aware of the current problematic situation of these families, and we have added more remarks regarding the Eoobolidae in the Systematic Palaeontology section of the manuscript. However, the revision of the families Eoobolidae and Lingulellotretidae falls outside the scope of this paper. We prefer to exclude it just now, as it will be part of an upcoming publication based on more material from China, Australia, Sweden and Estonia that we are currently working on.